# Clinical response to azacitidine in MDS is associated with distinct DNA methylation changes in HSPCs

Hypomethylating agents are frontline therapies for myelodysplastic neoplasms (MDS), yet clinical responses remain unpredictable. We conducted a phase 2 trial comparing injectable and oral azacitidine (AZA) administered over one or three weeks per four-week cycle, with the primary objective of investigating whether response is linked to in vivo drug incorporation or DNA hypomethylation. Our findings show that injection results in higher drug incorporation, but lower DNA demethylation per cycle, while global DNA methylation levels in mononuclear cells are comparable between responders and non-responders. However, hematopoietic stem and progenitor cells (HSPCs) from responders exhibit distinct baseline and early treatment-induced CpG methylation changes at regulatory regions linked to tissue patterning, cell migration, and myeloid differentiation. By cycle six–when clinical responses typically emerge–further differential hypomethylation in responder HSPCs suggests marrow adaptation as a driver of improved hematopoiesis. These findings indicate that intrinsic baseline and early drug-induced epigenetic differences in HSPCs may underlie the variable clinical response to AZA in MDS.

Myelodysplastic neoplasms (MDS) and chronic myelomonocytic leukaemia (CMML) are clonal malignancies driven by accumulation of somatic mutations in haematopoietic stem cells (HSCs), leading to impaired haematopoiesis and cytopenias along with increased likelihood of AML progression[1-6]. High risk patients who are ineligible for haematopoietic stem cell transplant are treated with hypomethylating agents (HMAs), including azacitidine (AZA), which in a subset of patients leads to improved peripheral cell counts and delayed progression to AML[7-10]. In the context of MDS/CMML, AZA is generally administered by subcutaneous injection (Vidaza®, 75 mg/m²/day; 7/28-day cycle). Clinical response is generally not apparent until after 4–6 treatment cycles[7], and for patients with primary or secondary resistance to HMAs, treatment options are limited to enrolment in relevant clinical trials or supportive care. Recently, an oral formulation of AZA (CC-486/Onureg®) has been approved for maintenance therapy in AML patients[11].

HMAs such as AZA and decitabine (DAC) are nucleoside analogues that undergo intracellular conversion[12,13] and are incorporated into DNA (and for AZA, into RNA[14]). HMA incorporation into DNA is cell cycle dependent[15], and we have previously reported a correlation between cell cycle parameters at diagnosis and subsequent patient response[16]. Following DNA incorporation, HMAs trap the maintenance DNA methyltransferase DNMT1, leading to formation of DNA-protein adducts, degradation and cellular depletion of DNMT1, and subsequent global hypomethylation[17-20]. Multiple stem-cell-intrinsic effects of HMAs observed in patients have been proposed to mediate the clinical effects of HMAs including epigenetic reactivation of tumour suppressor genes[21,22], activation of silenced transposable elements (TEs) and subsequent viral mimicry response[23-27], DNA-damage response and related cytotoxicity[17,18,28-32], induction of apoptosis[33-35], and RNA-dependent mechanisms[13,36-38]. However, HMA-mediated cytotoxicity is not essential for clinical response[39,40]. Indeed, in MDS/CMML, response to HMAs does not require eradication of mutated clones; rather, improved peripheral blood counts are driven by increased output from mutated stem cells[16,41-43]. Overall, no

✉e-mail: j.thoms@unsw.edu.au; mark.polizzotto@anu.edu.au; jpimanda@unsw.edu.au

straightforward relationship has been demonstrated between stem-cell-intrinsic effects of HMA treatment and patient outcome[16,21,44–47], rendering prediction of response status and development of rationally designed combination therapies an ongoing challenge.

Importantly, the association between HMA incorporation and DNA hypomethylation, and the clinical efficacy of these drugs, is unclear[48,49]. In a small retrospective study, HMA incorporation in a single initial treatment cycle tracked with global demethylation, with more variable uptake and hypomethylation observed in patients who did not subsequently respond[47]. However, HMA incorporation and DNA hypomethylation kinetics have not previously been studied over the time span where clinical response becomes evident. Furthermore, these parameters have not been assessed in parallel with cell cycle characteristics or changes in clonal composition during HMA treatment. To address these relationships, we prospectively collected bone marrow (BM) and peripheral blood (PB) as part of a phase II clinical trial (NCT03493646) designed to evaluate in vivo AZA incorporation in mononuclear cells following treatment with Vidaza (injection AZA) or CC-486 (oral AZA). The primary trial objective was to assess drug incorporation into DNA and global changes in DNA methylation using a liquid chromatography-tandem mass spectrometry (LC-MS/MS) based method[47], while secondary objectives included observation of cell cycle changes in haematopoietic stem and progenitor cells (HSPCs), tracking of clonal variants, and assessment of any relationship between these measures and clinical response.

## Results

### Patient cohort and clinical outcomes

Patients newly diagnosed with high risk (HR) MDS, CMML, or low blast acute myeloid leukaemia (AML), and ineligible for allogeneic stem cell transplant or intensive chemotherapy (Supplementary Table 1), were enroled to an open label phase 2 multicentre investigator-initiated trial (NCT03493646) and administered six cycles of injection AZA followed by six cycles of oral AZA (Fig. 1a). Clinical response was assessed following cycle (C) 6 day (D) 28 (C6D28) and cycle 12 day 28 (C12D28) using IWG2006 criteria[50] (Supplementary Table 2). Forty patients commenced treatment. Of the 24 who completed 6 cycles of injection AZA, there were 16 responders (seven complete remission [CR], three marrow complete response [mCR], and six haematological improvement [HI]) and eight non-responders (five stable disease [SD], three progressive disease [PD]/failures) (Fig. 1b, c, Supplementary Table 3). Participants were retrospectively assessed using IWG2023 criteria[51]; one patient (P10) changed from non-responder to responder at C12D28 (Supplementary Table 4).

Twenty-two patients entered the oral AZA phase and completed median 2 cycles (range 0–6); in total, 100 cycles of oral AZA were delivered. Reasons for discontinuation during the oral AZA phase (n = 11) included disease progression (five patients, 46%), unacceptable toxicity (three patients, 27%; one haematological, two gastrointestinal), changes in goals of care (two patients, 18%), and intercurrent medical complication (one patient, 9%). The most common adverse events during oral AZA therapy were haematological (Supplementary Tables 5, 6, 7, and 8): 23 cycles in 13 patients were complicated by grade 4 neutropenia, one cycle in one patient by febrile neutropenia, and 14 cycles in 10 patients by grade 4 thrombocytopenia. Other adverse events were less common, with 13 cycles in nine patients complicated by grade 3 gastrointestinal toxicity (diarrhoea; five events in three participants, nausea; four events in three participants, rectal haemorrhage; two events in one participant, one event each for vomiting and abdominal pain; no grade 4 adverse events were observed).

11/22 patients who commenced oral AZA reached the second response assessment (C12D28) comprising seven responders (two CR, four mCR, one HI) and four non-responders (two SD, one PD, one relapse after CR/PR) (Supplementary Table 3). The seven responders at C12D28 included five responders and two non-responders at C6D28 (i.e. two patients with a delayed clinical response), while the four non-responders at C12D28 had previously shown response at C6D28 (Fig. 1c). Most patients completing cycle 12 elected to continue to receive treatment with oral AZA (seven patients, 64%).

### Cell cycle parameters in CD34+ HSPCs at diagnosis and over the course of treatment do not correlate directly with clinical outcome

The proportion of CD34+ HSPCs in the BM-MNC fraction was slightly elevated in non-responders (Supplementary Fig. 1a), and increased bone marrow trephine (BMT) cellularity was associated with a higher proportion of non-responders (Supplementary Fig. 1b and Supplementary Table 9). Previous studies suggested that patients who respond to AZA have a higher proportion of actively cycling (S/$G_2$/M phase) hematopoietic progenitor cells (HPC) at baseline compared to patients who do not have a clinical response[16]. RNAseq analysis of patients prior to AZA treatment showed gene expression patterns reminiscent of an earlier cohort (Supplementary Fig. 2a), with relative enrichment of genes expressed in dividing CD34$^+$ cells (Supplementary Fig. 2b). We measured cell cycle parameters in bone marrow CD34$^+$CD38$^{lo}$ HSC and CD34$^+$CD38$^+$ HPC across the course of AZA treatment (Fig. 2a and Supplementary Fig. 1c, d). In HSCs, there were few actively cycling cells at baseline regardless of subsequent clinical response (Fig. 2b, median % in cell cycle phase, [interquartile range]; Responders (R): [0.5, (0.1–1.2)]; Non-responders (NR): [0.2, (0.1–0.5)]), and the overall proportion of cells in $G_0$ or $G_1$ were similar between response groups (Fig. 2c, $G_0$ - R: [83.2, (68.7–90.7)]; NR: [57.7, (48.6–91.4)], Fig. 2d, $G_1$ - R: [16.6, (6.3–25.6)]; NR: [35.3, (7.4–44.6)]). Following six cycles of injection AZA the proportion of HSCs in S/$G_2$/M was relatively stable (Fig. 2b, R: [0.5, (0.0–0.9)]; NR: [1.2, (0.6–1.4)]). There was an overall decrease in the proportion of quiescent $G_0$ cells (Fig. 2c, R: [53.8, (47.2–60.8)]; NR: [56.8, (44.6–61.2)]) and concomitant increase in the proportion of cells in $G_1$ (Fig. 2d, R: [42.6, (37.5–48.5)]; NR: [40.9, (35.0-53.4)]), which was significant in both response groups. Although RNA sequencing data suggested cell cycle differences between response groups at baseline (Supplementary Fig. 2a, b), the proportion of HPCs in S/$G_2$/M at baseline were similar (Fig. 2e; R: [5.0, (2.9–5.7)]; NR: [3.2, (2.2–6.0)]), and a published 20-gene signature[16] did not dichotomise patient outcome in this cohort (Supplementary Fig. 2c). Similar to HSCs, the overall proportion of $G_0$ and $G_1$ HPCs were similar between response groups at baseline (Fig. 2f, $G_0$ - R: [59.9, (55.8–69.8)]; NR: [48.4, (29.8–71.2)]; Fig. 2g, $G_1$ - R: [32.9, (25.0–38.6)]; NR: [48.5, (24.0–1.5)]). After six treatment cycles, some patients had shifts in the proportion of HPCs in S/$G_2$/M, but there was no consistent change either across the entire cohort, or within response groups (Fig. 2e, R: [7.2, (4.1–9.9)]; NR: [7.3, (5.7–9.2)]). However, similar to HSCs there was an overall decrease in the proportion of quiescent HPCs (Fig. 2f, $G_0$ - R: [23.5, (17.3–38.7)]; NR: [35.2, (24.2–40.4)]) and an increase in the proportion of $G_1$ cells (Fig. 2g, R: [69.4, (56.4–72.8)]; NR: [57.5, (54.4–66.1)]) which were significant in both response groups.

At the start of the oral phase (C7D1), expression of cell division-associated genes[52] in CD34$^+$ cells was enriched in responders compared to non-responders (Supplementary Fig. 2d). HSC cell cycle parameters were similar between responders and non-responders (Supplementary Fig. 2e, S/$G_2$/M - R: [0.6, (0.3–1.1)]; NR: [0.7, (0.3–1.3)]; Supplementary Fig. 2f, $G_0$ - R: [47.9, (38.5–58.5)]; NR: [53.8, (47.3–60.6)]; Supplementary Fig. 2g, G1 - R: [43.7, (39.4–59.4)]; NR: [43.2, (37.1–51.5)]). Cell cycle parameters in HPCs at the start of the oral phase were also similar between response groups (Supplementary Fig. 2h, S/$G_2$/M - R: [5.4, (4.3–7.3)]; NR: [8.2, (4.2–9.9)]; Supplementary Fig. 2i, $G_0$ - R: [21.6, (17.0–28.6)]; NR: [39.5, (17.4–41.1)], Supplementary Fig. 2j, G1 - R: [72.2, (67.1–73.6)]; NR: [57.8, (52.1–72.6)]). Responder patients showed an increase, and non-responder patients a decrease in the proportion of cells in S/$G_2$/M over the oral phase (Supplementary

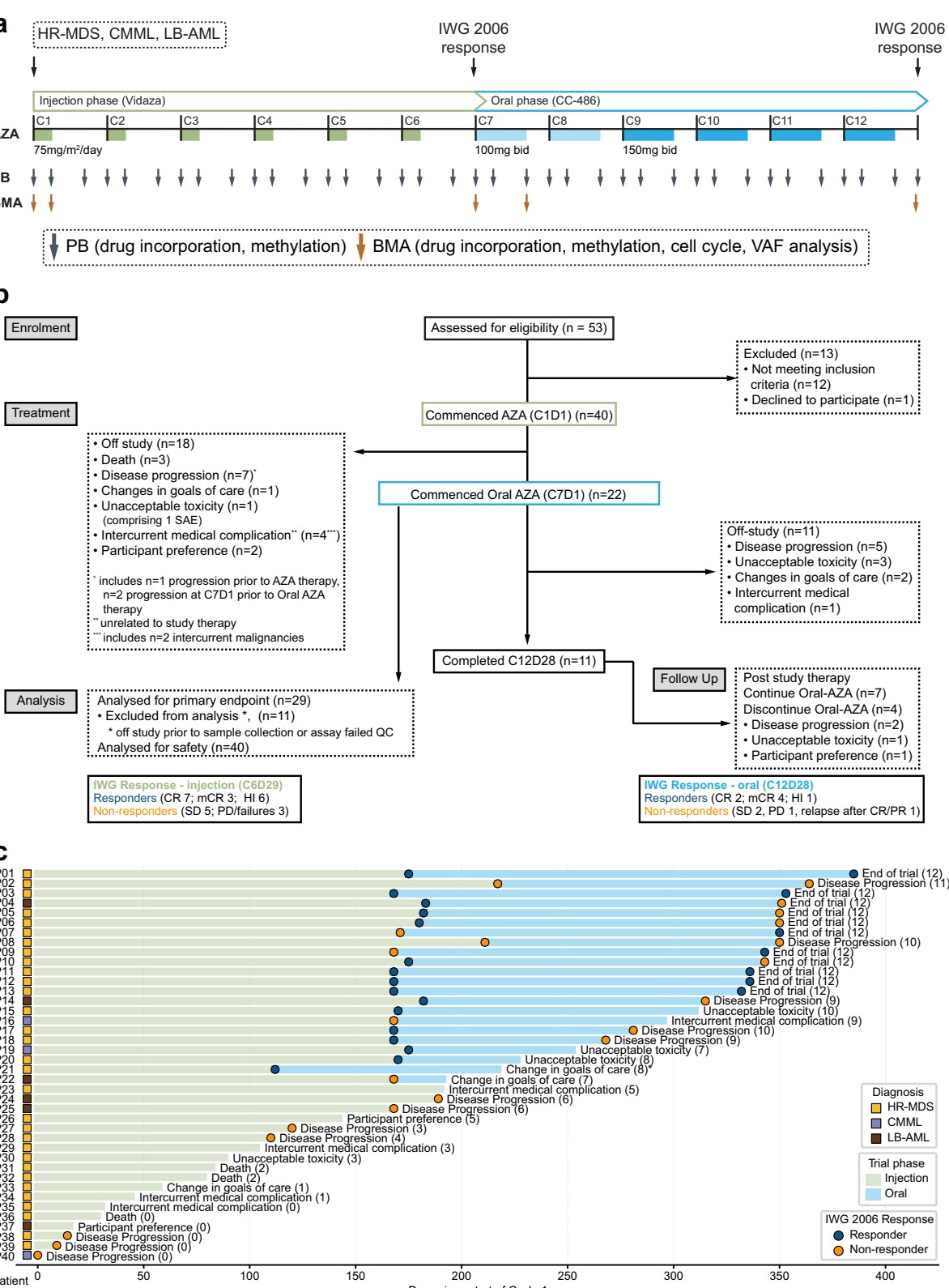

**Fig. 1 | Trial design and patient outcomes. a** Trial design showing dosing schedule, specimen collection, and IWG assessment timepoints for injected and oral phases. **b** Consort diagram summarising outcomes for all participants who were assessed for eligibility. **c** Swimmer plots showing participant response and outcome over time. Coloured squares indicate diagnosis for each patient (yellow−high risk (HR) MDS, purple−CMML, brown−low blast (LB) AML). Circles show response at IWG assessment or progression timepoints (blue−responder, orange−non-responder). The off-study reason for each patient is as indicated, the number in brackets indicates the number of treatment cycles completed, * denotes a single patient (P21) who was accelerated to the oral phase following 4 injection cycles.

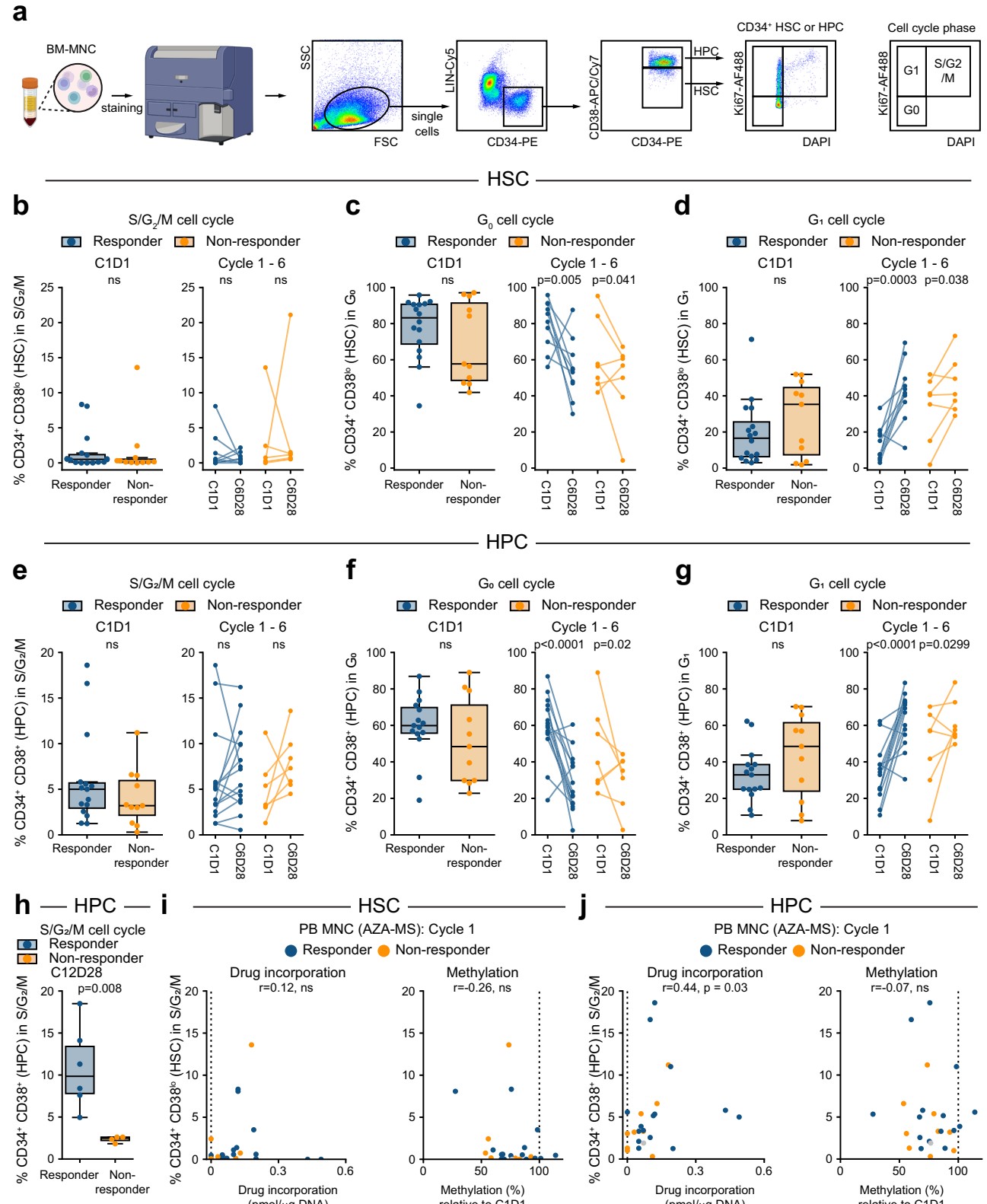

Fig. 2h). Strikingly, by the end of treatment, HPCs in non-responder patients had essentially stopped cycling (Fig. 2h, R: [9.9, (7.8–13.4)]; NR: [2.5, (2.2–2.6)]).

Cell cycle kinetics might directly influence DAC incorporated into DNA, and therefore the extent of DNA demethylation. We compared cell cycle parameters with DAC incorporation and global DNA demethylation in circulating cells during cycle 1 in HSCs (Fig. 2i) and HPCs

(Fig. 2j). Immediately following a single treatment cycle, DAC incorporation and DNA demethylation were observed in essentially all patients, and the degree of DAC incorporation was correlated to the proportion of HPCs in S/G2/M (Fig. 2j; Spearman r = 0.44, *P* = 0.03). During cycle 7 (oral phase) DAC incorporation and demethylation were again observed in essentially all patients, however, neither correlated with cell cycle parameters at this time point (Supplementary Fig. 2k, l).

**Fig. 2 | Cell cycle parameters in CD34+ HSPCs at diagnosis and across treatment cycles. a** Analysis workflow for cell cycle analysis in CD34$^+$ CD38$^{lo}$ stem (HSC) and CD34$^+$ CD38$^{hi}$ progenitor (HPC) cells. Schematic partially created in BioRender. Thoms, J. (2025) https://BioRender.com/64wue88. Percentage of HSCs in **b** S/G$_2$/M, **c** G$_0$, **d** G$_1$ cell cycle phases. Samples with <50 immunophenotypic HSCs not shown. **b–d** Left: percentage of cells in specified cell cycle phase at diagnosis (C1D1), coloured by outcome at end of the injection phase. Responders: $n = 15$, non-responders $n = 11$. $P$ values are indicated, ns denotes $P > 0.05$, Welch's two-sided $t$-test. Boxplots throughout this figure show centre=median, box=interquartile range (IQR), whiskers=furthest point within 1.5× IQR, outliers = points >1.5× IQR. Right: Percentage of cells in specified cell cycle phase at diagnosis (C1D1) and end (C6D28) of injection phase. Lines indicate paired samples from a single patient, only patients with data at both timepoints are shown. Responders: $n = 10$, non-responders $n = 7$. $P$ values as indicated, ns denotes $P > 0.05$, two-sided linear mixed model. Percentage of HPCs in (**e**) S/G$_2$/M, (**f**) G$_0$, (**g**) G$_1$ cell cycle phases. **e–g** Left: percentage of cells in specified cell cycle phase at diagnosis (C1D1), coloured by outcome at end of the injection phase. Responders: $n = 15$, non-responders $n = 11$. $P$ values are indicated, ns denotes $P > 0.05$, Welch's two-sided $t$-test. Right: Percentage of cells in specified cell cycle phase at diagnosis (C1D1) and at the end (C6D28) of the injection phase. Lines indicate paired samples from a single patient, only patients with data at both timepoints are shown. Responders: $n = 15$, non-responder $n = 7$. $P$ values as indicated, ns denotes $P > 0.05$, two-sided linear mixed model. **h** Percentage of HPCs in S/G$_2$/M following 12 treatment cycles, coloured by clinical outcome at C12D28, Welch's two-sided $t$-test. Relationship between cell cycle status and drug incorporation and DNA demethylation during cycle 1 in (**i**) HSCs and (**j**) HPCs. **i, j** Percentage of cells actively cycling (S/G$_2$/M phase) at C1D1 compared to maximum drug incorporation (left) and minimum DNA methylation (right) in peripheral blood during the same cycle. Dashed lines indicate baseline values, grey dots show patients with no response data. r = two-sided spearman correlation coefficient.

Overall, we found that most patients had fewer quiescent cells following six cycles of AZA treatment, and at the end of cycle 12, HPCs in responder patients continued to cycle, while HPCs in non-responders had exited the cell cycle. However, cell cycle parameters did not directly correlate with clinical outcome.

## DAC incorporation and global DNA demethylation in PB and BM MNCs are not directly correlated with clinical response

AZA undergoes intracellular modification prior to cell cycle dependent DNA incorporation and subsequent DNA hypomethylation[19] (Fig. 3a). We directly measured drug incorporation into DNA and relative global DNA methylation in PB mononuclear cells (MNCs) at 1–2 weekly intervals during both injection and oral treatment phases (Fig. 1a). During the injection AZA phase, drug incorporation was higher in responders [0.093 (95% CI 0.066–0.120)] compared to non-responders [0.045 (95% CI 0.011–0.079)] pmol DAC/µg DNA (Fig. 3b). Incorporation differences were primarily driven by day 8 (D8) measurements, i.e. immediately following 7 days of AZA treatment (Fig. 3b). However, overall differences in DNA methylation relative to baseline (100%) were comparable between responders [85.41% (95% CI 77.40–93.43)] and non-responders [86.85% (95% CI 76.69–97.01], with peak demethylation observed at D8 in both treatment groups (Fig. 3c and Supplementary Figs. 3a, b, c, d, and 4).

Nineteen patients completed at least one cycle of oral AZA at 150 mg bid (i.e., completed C9) and were compared to all patients with injection AZA data ($n = 29$) for assessment of the primary outcome. Drug incorporation was higher during the injection phase (Fig. 3d: injection AZA; 0.070 (95% CI 0.052–0.089) vs. oral AZA; 0.047 (95% CI 0.023–0.070) pmol DAC/µg DNA), again primarily driven by D8 measurements (Fig. 3d). However, there was lower overall DNA methylation during the oral phase (Fig. 3e: injection AZA; 86.42% (95% CI 81.06–91.78) vs. oral AZA; 77.64% (95% CI 71.67–83.61)). Methylation differences were significant at both day 1 (D1) and day 22 (D22), but not at D8 (Fig. 3e). Thus, although we observed increased drug incorporation with injection AZA, there was greater and more sustained demethylation in response to oral administration of AZA.

During the oral phase, drug incorporation was comparable between response groups (Fig. 3f: responders; 0.059 (95% CI 0.012–0.106) vs. non-responders; AZA 0.073 (95% CI 0.031–0.115) pmol DAC/µg DNA). Similar to injection AZA, we did not observe any methylation differences between the response groups (Fig. 3g: responders; 79.61% (95% CI 62.35–96.87) vs. non-responders; 79.23% (95% CI 64.31–94.15)).

Although DNA demethylation is a likely mediator of the therapeutic effects of AZA, in this cohort essentially every patient had reduced DNA methylation in peripheral blood following treatment, and drug incorporation, level of global DNA demethylation, and clinical outcome were not directly related (Fig. 3c, g). One possibility is that the clinically relevant demethylation events occur in specific cells

within the bone marrow and may not be read out in circulating cells. We therefore measured global DNA demethylation in BM MNCs before and after AZA treatment in cycle 1 and cycle 7. Similar to PB, demethylation was apparent in BM MNC from the majority of patients and was not significantly different between response groups (Fig. 3h).

MDS/CMML are diseases of stem cells, and clinical response in this cohort consists of improved circulating blood counts, likely driven by improved output from CD34$^+$ HSPCs[42]. HSPCs are rare and not amenable to AZA-MS analysis. However, relative methylation of LINE-1 promoters as a proxy measure of global methylation changes can be assessed in small cell numbers using a PCR-based assay[53] (Fig. 3i, Supplementary Fig. 3e). Comparing CD34$^+$ cells from responders and non-responders, average demethylation was greater in responders, most markedly at C7D22 (Fig. 3j). Taken together, our data indicated that most patients treated with AZA undergo drug incorporation and global DNA demethylation, and that the level of global demethylation in CD34$^+$ HSPCs may be related to clinical response.

## Pre-treatment DNA methylation at specific CpG sites in CD34+ HSPCs correlates with clinical response

To further investigate relationships between DNA methylation and clinical outcome we performed reduced representation bisulfite sequencing (RRBS) on longitudinal samples of BM HSPCs prior to, and immediately following, AZA treatment during cycle 1 and cycle 7 (Fig. 4a). At baseline (C1D1), the number of CpGs detected with more than 10 reads (range 2.4–3.6 million) and total reads in detected CpGs (range 38.6–114.9 million) were similar between responders and non-responders when looking at all samples (Supplementary Fig. 5a) or just MDS samples (Supplementary Fig. 5b), and hierarchical clustering showed highest similarity within samples derived from the same patient (Supplementary Fig. 5c). Global hypermethylation[54] is a feature of MDS/CMML/AML. However, at C1D1, we observed relative hypomethylation in HSPCs of responders compared to non-responders (Fig. 4b and Supplementary Fig. 5b), suggesting that differences in DNA methylation at diagnosis might influence patient outcomes. 23950 CpGs hypomethylated in responders compared to non-responders were predominantly located in CpG islands and promoter regions, while 3004 CpGs hypermethylated in responders were more frequently located in distal regions (Fig. 4c). Since cytosine methylation is often concordant in proximity, we combined differentially methylated cytosines (DMCs) to regions (DMRs) prior to clustering baseline data. Patients clustered by response group, with the average methylation percentage of DMRs remaining consistent within response groups, indicating shared regions of differential methylation between individuals (Fig. 4d). We then linked DMCs to target genes using HiChIP data from healthy HSPCs[55] and used pathway analysis to predict functional outcomes of differential methylation between responders and non-responders. Genes associated with hypomethylated DMCs in responders included *HOXA* and *HOXB* cluster genes,

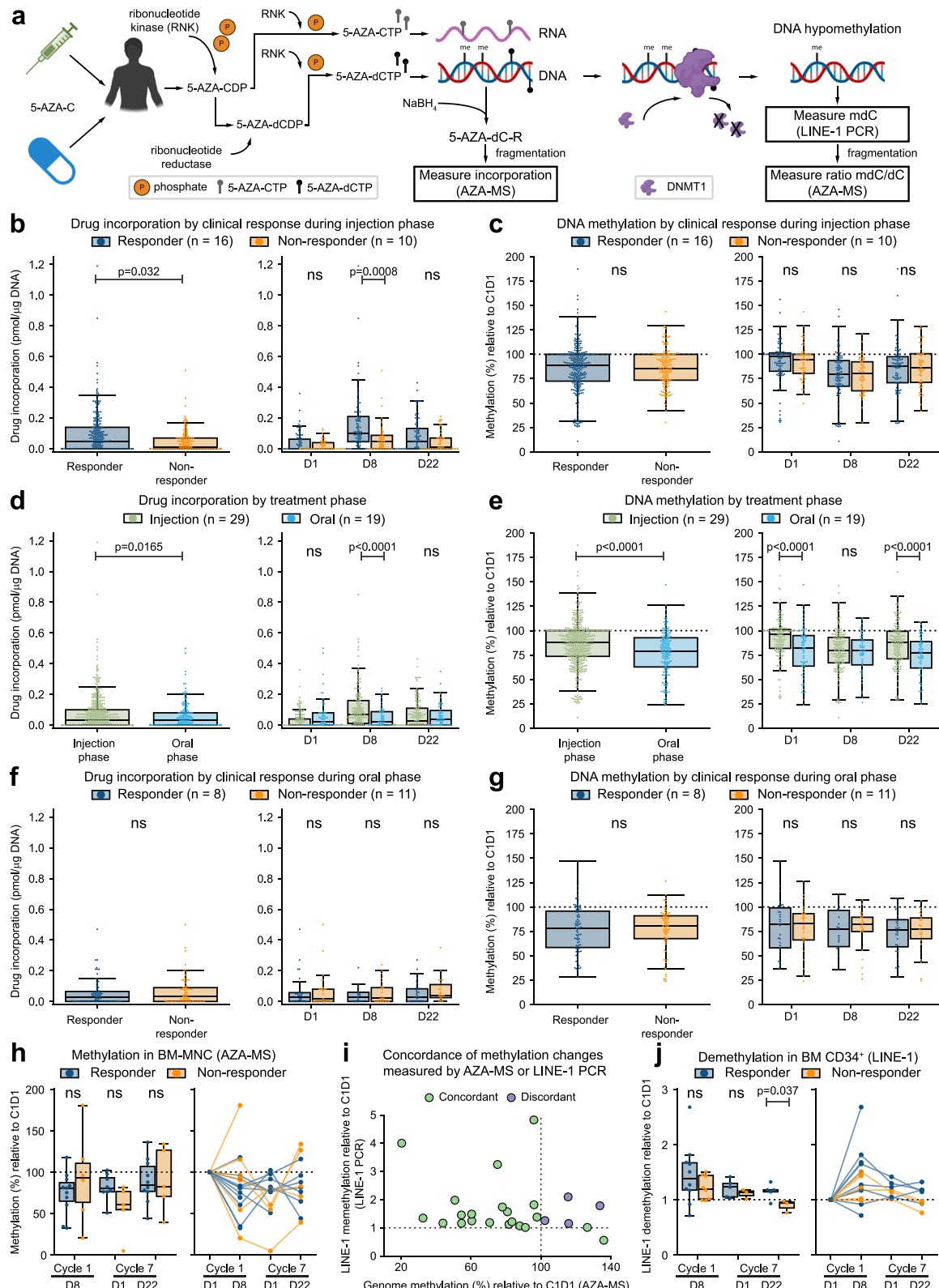

*GATA2*, *EPO*, and *SNAI1* (Fig. 4e, f). The top 10 gene ontology hits included pattern specification processes and regulation of epithelial cell migration (Supplementary Fig. 5d).

Baseline differences in CpG methylation suggest that CD34+ HSPCs might be epigenetically primed for AZA response in a subset of patients. To better understand how hypomethylated DMCs might influence the global epigenetic environment, we overlapped baseline DMCs with

global transcription factor (TF) and histone (H) chromatin immuno-precipitation sequencing (ChIPseq) data from healthy HSPC subsets (HSC-multipotent progenitors (MPP), common myeloid progenitor (CMP), granulocyte-monocyte progenitor (GMP), megakaryocyte-erythroid progenitor (MEP))[56]. Many regulatory elements used in healthy HSPCs are likely to have regulatory functions in MDS HSPCs, although these may be modulated by network rewiring in transformed

**Fig. 3 | DAC incorporation and relative DNA methylation in PB and BM MNCs across treatment phases. a** Administration of azacitidine (5-AZA-C), in vivo modification, and DNA incorporation leading to DNMT1 degradation and DNA hypomethylation. Drug incorporation into DNA is assessed by measuring NaBH$_4$-stabilised and fragmented DNA (dihydro 5-AZA-dC ribonucleotide (R)) by mass spectrometry. Schematic partially created in BioRender. Thoms, J. (2025) https://BioRender.com/6ep5ch6. **b, c** Drug incorporation and DNA methylation (relative to baseline) during injection phase (*n* = 26, responder: 16, non-responder: 10]). **b** Drug incorporation in responders and non-responders at all sample points (left) or separated by cycle day (right). Statistical comparisons for **b–g** were performed using a two-sided linear mixed model approach with subject-level random effects to account for within-subject variability, significant *P* values are as indicated, ns indicates *P* > 0.05. Boxplots throughout this figure show centre = median, box = interquartile range (IQR), whiskers = furthest point within 1.5xIQR, outliers = points >1.5xIQR. **c** Relative DNA methylation at all sample points or separated by cycle day. **d, e** Drug incorporation and DNA methylation during the injection and oral phases of the study (injection: *n* = 29, oral: *n* = 19) (**d**) DAC incorporation during the injection and oral phase at all sample points (left) or separated by cycle day (right). **e** Relative DNA methylation at all sample points or separated by cycle day. **f, g** Drug incorporation and DNA methylation during oral phase of the study (*n* = 19, responder: 8, non-responder: 11). **f** DAC incorporation in responders and non-responders at all sample points (left) or separated by cycle day (right). **g** Relative DNA methylation at all sample points or separated by cycle day. (**h**) Relative DNA methylation in bone marrow (BM) mononuclear cells (MNC) compared to pre-treatment (*n* = 17, responder: 11, non-responder: 6). Left: aggregate data at each timepoint. Right: Longitudinal methylation changes. Statistical comparisons for (**h, j**) were performed using two-sided *t*-test_ind (scipy.stats). **i** Comparison of methylation changes measured by mass spectrometry or LINE-1 PCR assay in BM MNC. Green–concordant methylation changes in both assays. Purple–methylation changes differ between assays (*n* = 25). **j** Relative LINE-1 DNA demethylation in BM CD34$^+$ cells compared to pre-treatment (*n* = 17, responder: 10, non-responder: 8). Left: aggregate data at each timepoint. Right: Longitudinal methylation changes.

cells[57]. There was striking overlap between TF/histone binding and DMCs hypomethylated in responders, but only at select histone marks for DMCs hypermethylated in responders (Fig. 4g). Both hypo- and hyper-methylated DMCs showed overlap with H3K4me3 (trimethylated lysine 4; promoter mark) and H3K27me3 (trimethylated lysine 27; repressive mark), which together mark bivalent regions primed for epigenetic plasticity[58,59]. CTCF is involved in defining chromatin boundaries, and along with STAG2 belongs to the cohesin complex, which facilitates looping of promoters to distal regulatory regions[60,61]. We observed enriched overlap with CTCF and STAG2 binding sites only at DMCs that were hypomethylated in responders (Fig. 4g). Finally, overlap with TF binding sites was most prominent at FLI1-bound regions in CMPs and MEPs, and RUNX1-bound regions in CMPs. In CMPs and MEPs both FLI1 and RUNX1 bind regulatory regions of lineage-specific genes that are subsequently expressed in mature myeloid or erythroid cells[55] which are the specific cell populations that contribute to clinical response.

Despite significant differences in CpG methylation, global gene expression profiles were similar between response groups at C1D1 (Fig. 4h and Supplementary Fig. 5j). Given that FLI1 and RUNX1 are involved in lineage decisions, and clinical response requires improved production of circulating cells, we focussed on gene pathways associated with blood cell differentiation. At C1D1, we observed minimal enrichment of pathways regulating myeloid and erythroid differentiation (Fig. 4i), consistent with all patients being unwell at this timepoint. Overall, we observe significant baseline CpG hypomethylation in patients who go on to respond to AZA, with the hypomethylated regions overlapping genes critical for blood development, and regulatory sites associated with epigenetic plasticity and lineage-specific gene regulation.

### Longitudinal assessment of DNA methylation in CD34+ HSPCs shows that dynamic changes at specific CpG sites correlate with clinical response

We next looked at methylation changes immediately following initial AZA treatment. Extensive hypomethylation was evident across all patients, with more than twice as many CpGs hypomethylated in responders compared to non-responders (Fig. 5a; 23060 vs 9580). Hypermethylation compared to baseline was also observed at a very small number of CpGs (Fig. 5a; 79 vs 40). While baseline methylation differences were mostly promoter-proximal, hypomethylation following AZA treatment was mostly at distal or intronic regions which potentially correspond to enhancers (Fig. 5a). Comparing overlap of specific DMCs between responders and non-responders, 7572 CpGs were hypomethylated in both responders and non-responders; these comprised the majority of DMCs in non-responders (Fig. 5b). Gene ontology analysis of genes associated with the shared hypomethylated CpGs revealed a single enriched pathway (Supplementary Fig. 5e;

immune response regulating signalling pathway). A further 15488 CpGs were uniquely hypomethylated in responders, while 2005 CpGs were uniquely hypomethylated in non-responders. Genes associated with CpGs uniquely hypomethylated in responders were enriched for pathways relating to myeloid cell and osteoclast differentiation (Fig. 5c and Supplementary Fig. 5f), suggesting clinical response to AZA is at least partly due to epigenetic reprogramming at gene loci required for HSPCs to differentiate and produce circulating progeny. Overlap of hypomethylated CpGs in responders at C1D8 with global transcription factor (TF) and histone ChIPseq data from healthy HSPC subsets revealed enrichment at regulatory regions, particularly at H3K27ac sites (lysine 27 acetylated; active enhancers) in GMPs and MEPs, again consistent with improved differentiation capacity (Fig. 5d). Overall gene expression changes at C1D8 were dominated by cell cycle pathways, an expected effect of AZA treatment (Fig. 5e), and response-specific enrichment of pathways associated with myeloid and erythroid differentiation which were shifted compared to C1D1 (Fig. 5f).

Clinical response to AZA is generally not evident for several months following commencement of treatment[7], and the initial changes in CpG methylation and gene expression may not persist over multiple treatment cycles. We therefore compared CpG methylation at C7D1 (after 6 AZA treatment cycles, but 3 weeks after the last AZA dose) to baseline methylation. Hypomethylation relative to baseline was observed in all patients, again skewed to distal regulatory regions, with approximately three times as many sites hypomethylated in responders compared to non-responders (Supplementary Fig. 6a; 889 vs 315). However, the overall number of hypomethylated sites was significantly reduced compared to C1D8, and there was no overlap in hypomethylated sites between response groups (Supplementary Fig. 6b). Gene ontology analysis of genes associated with hypomethylated CpGs in responders revealed enrichment in multiple pathways including regulation of cell-cell adhesion and positive regulation of leucocyte differentiation and hemopoiesis; these pathways may reflect a marrow environment that is more supportive of blood production (Supplementary Figs. 6c and 5g). Consistent with flow cytometry data, global gene expression changes were dominated by cell cycle-associated pathways (Supplementary Fig. 6d). Comparing gene expression between response groups revealed enrichment of immune-associated pathways in responders, while non-responders were enriched for extracellular matrix- and erythrocyte-associated pathways (Supplementary Fig. 6e, f, g).

C7D1 samples are collected following 21 days without AZA treatment and thus represent an altered baseline state where patient response is known. Similar to C1D1, we observed relative hypomethylation in CD34$^+$ cells of responders compared to non-responders (Supplementary Fig. 7a, b), with 19650 CpGs hypomethylated in responders and predominantly located in promoter regions. However, more than half as many CpGs (13034) were hypomethylated in non-

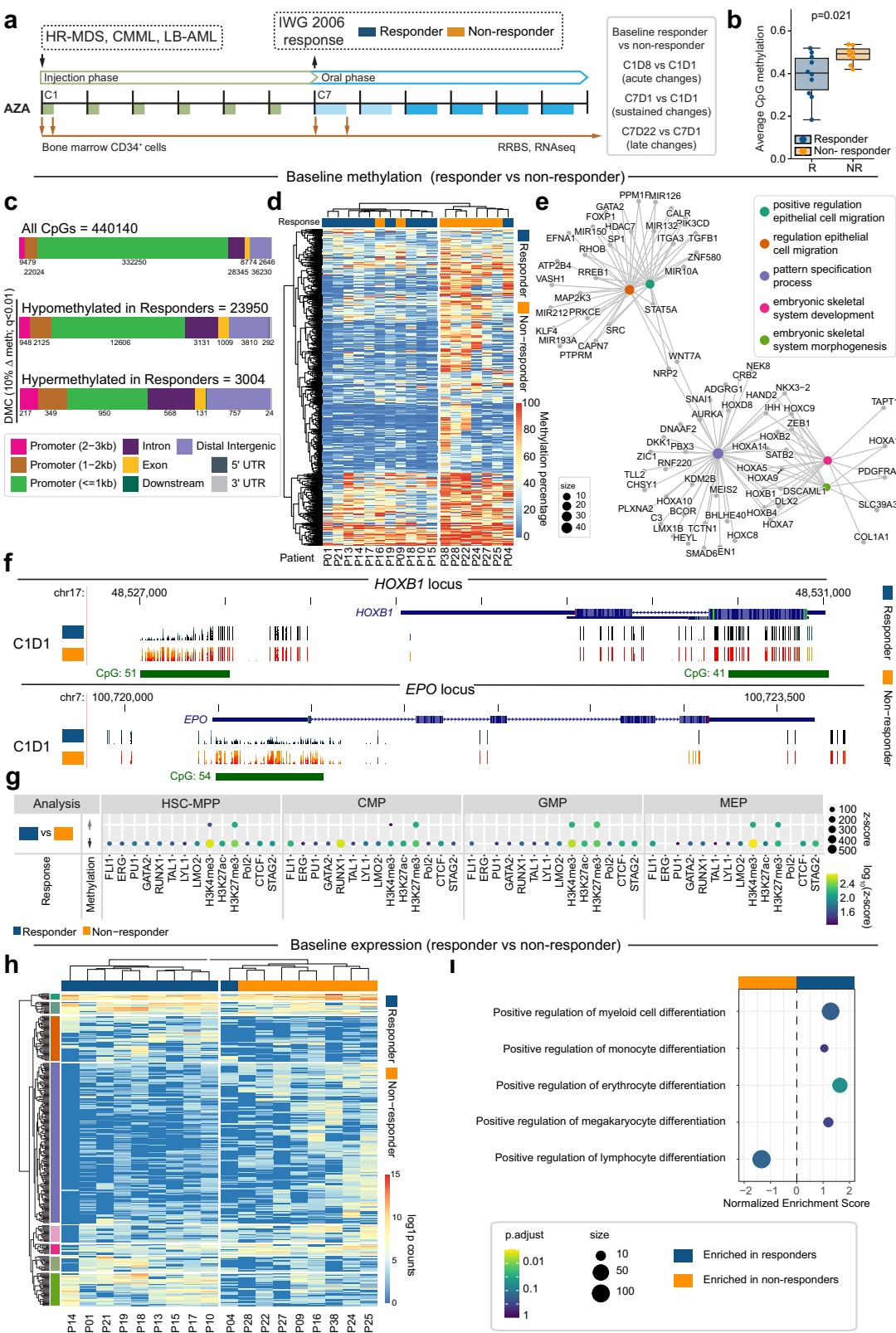

responders and were more frequently located in distal regions (Supplementary Fig. 7a). Hypomethylated DMCs in responders mapped to pathways suggestive of alterations in developmental regulation and immune function (Supplementary Figs. 7c and 5h). Consistent with improved circulating output in responders, gene expression changes showed significant enrichment of pathways related to myeloid and erythroid cell differentiation (Supplementary Fig. 7d).

Finally, to assess changes after long term AZA treatment we compared CpG methylation at C7D22 to C7D1. Surprisingly, we observed mostly hypermethylation in all patients, albeit at a very small number of CpGs (Supplementary Fig. 8a, b; responders - 106, non-responders - 83). Pathway analysis of genes associated with hypermethylated CpGs returned no hits. Eighteen CpGs were hypomethylated in responders (Supplementary Fig. 8a, b), and associated

**Fig. 4 | Baseline differences in site-specific CpG methylation in CD34+ HSPCs correlate with clinical response to AZA. a** Schematic showing sample collection timepoints for reduced representation bisulphite sequencing (RRBS). **b** Average methylation at CpG sites at baseline (C1D1) for $n = 10$ responders and $n = 8$ non-responders. Statistical comparison is unpaired two-sided Welch's $t$-test. Boxplots show centre=median, box=interquartile range (IQR), whiskers = furthest point within 1.5xIQR, outliers = points >1.5xIQR. **c–i** Comparison of responder ($n = 10$) and non-responder ($n = 8$) patients prior to commencing AZA therapy. **c** Upper bar shows genomic distribution of 440140 CpG sites with data in all samples at C1D1 and C1D8. Lower bars show genomic distribution of CpGs which were hypo-methylated ($n = 23950$) or hypermethylated ($n = 3004$) in responders compared to non-responders. **d** Clustered heatmap showing differentially methylated regions across all patients, with clinical response as indicated. **e** Network diagram showing enriched pathways for genes mapping to CpGs which are hypomethylated in responders at baseline. Network diagrams were created with clusterProfiler[91] using

GO pathways[98]. CpGs were annotated to genes using HiChIP data from healthy human HSPC subsets[55]. **f** Representative UCSC tracks showing differentially methylated CpGs at the *HOXB1* (chr17:48,526,692-48,531,309 [hg38]) and *EPO* (chr7:100,719,795-100,724,032 [hg38]) loci. Tracks show composite data for each response group at C1D1. **g** Enrichment of differentially methylated CpGs at genomic regions with specific histone marks (H3K27Ac, H3K3me3, H3K27me3) or bound by key transcription factors and genome organisers (FLI1, ERG, PU.1, GATA2, RUNX1, TAL1, LYL1, LMO2, Pol2, CTCF, STAG2) in healthy human HSPC subsets. Plot shows CpG regions hypermethylated (upper) or hypomethylated (lower) in responders compared to non-responders. **h** Clustered heatmap showing log(1+count) for genes differentially expressed between responders and non-responders at C1D1, with clinical response as indicated. Cluster colours correspond to pathways shown in Supplementary Fig. 5j. **i** Normalised enrichment scores from FSGEA[99] analysis for pathways related to blood production in responders compared to non-responders at C1D1.

genes were enriched in pathways including regulation of signal transduction by p53 class mediator and regulation of mitotic cell cycle phase transition (Supplementary Figs. 8c and 5i). In contrast to earlier timepoints, overall expression changes mapped to chromatin organisation and GTPase cycle (Supplementary Fig. 8d) with responders again enriched for pathways related to myeloid differentiation (Supplementary Fig. 8e).

Altered CpG methylation patterns could potentially impact RNA splicing[62,63] and indeed, altered splicing is a general feature of MDS[64–66]. We therefore investigated differential transcript expression (DTE: Supplementary Fig. 9a) and differential transcript usage (DTU: Supplementary Fig. 9b). DTE events were most prominent in responders at C7D22 v C7D1, while DTU events were observed at C7D22 v C7D1 and at baseline. However, there was modest overlap between genes with DTE or DTU events and pathways related to blood differentiation. DTE (and to some extent DTU) events also overlapped genes with differential methylation (Supplementary Fig. 9c), including the platelet gene *NBEAL2* and splicing factor *SRSF5*. Dysregulation of RNA splicing could also result in production of novel RNAs via intron retention and altered exon usage. At baseline, we observed abundant skipped exon events (SE) with a slightly higher fraction occurring in responders. Retained intron (RI) events were less abundant, but highly skewed to non-responders. Gene ontology analysis of SE and RI transcripts identified the same enriched pathways in responders and non-responders; these were overwhelmingly RNA splicing and RNA processing related pathways (Supplementary Fig. 9d). At C7D1, there were fewer unique RI events, and the imbalance between responders and non-responders had resolved. However, there were striking differences in the pathways associated with RI events, with mostly viral response-related pathways enriched in responders, but RNA processing pathways enriched in non-responders (Supplementary Fig. 9e). Pathways associated with SE events remained similar between responders and non-responders, although these were not the same pathways observed at baseline and might represent universal consequences of HMA treatment.

Expression of TEs has been suggested to mediate HMA efficacy by activating inflammatory responses via "viral mimicry"[19,23,25]. Similar to genome-wide results (Fig. 4b) and LINE-1 PCR data (Fig. 3j), CpGs associated with either long interspersed nuclear element (LINE), long terminal repeat (LTR), or short interspersed nuclear element (SINE) families of TEs were less methylated at C1D1 in patients who went on to respond to AZA (Fig. 6a), and average CpG methylation at major TE families was lower at C1D8 compared to C1D1 in both response groups (Fig. 6b). However, methylation differences did not translate to response-specific differences in TE expression at C1D8 (Fig. 6c). TE methylation returned to baseline levels by C7D1 and was not sub-stantially changed at C7D22 (Fig. 6d, e). However, there was an overall pattern of increased TE expression in responders at C7D22 (Fig. 6f) which mapped to 12 differentially expressed TE subfamilies, mostly belonging to L1PA# sub-families (Fig. 6g). Consistent with TE

expression potentially driving a viral mimicry response, we also observed enrichment of inflammatory response pathways in responders at C7D22 (Fig. 6h).

## Variant alleles in BM MNCs persist over the course of AZA treatment irrespective of clinical response and demethylation kinetics

CD34+ HSPCs are the origin of improved peripheral blood counts in responders. An ongoing question has been whether AZA treatment clears mutated HSPC clones from the marrow, allowing expansion of residual healthy stem cells, or conversely, reprogrammes mutated cells such that blood output is increased. We tracked mutations in 111 genes associated with myeloid pathologies (Supplementary Table 10) during of AZA treatment (Supplementary Data 1). Our cohort had baseline mutational profiles typical of MDS/CMML/AML[67] with the most frequent mutations occurring in *TET2* (10/28; 35.7%), *RUNX1* (10/28; 35.7%), *SRSF2* (9/28, 32.1%), and *TP53* (8/28, 28.6%) (Supplementary Fig. 10a, b).

Most patients had stable sub-clonal composition across treatment phases, irrespective of clinical outcome (Fig. 7a). Three patients showed substantially reduced mutational burden corresponding to clinical response at C6D28 with subsequent reappearance of the mutated clone at progression (Fig. 7a; blue arrows). Conversely, two patients were initially non-responders but showed a delayed response; in both cases mutational burden was initially stable but reduced at the time of response (Fig. 7a; orange arrows). We observed patients with sustained response but fluctuating mutational burden (Fig. 7b; P01), patients with stable mutational burden across initial response and subsequent progression (Fig. 7b; P05), and patients with decreasing mutational burden in the absence of clinical response (Fig. 7b; P08). Overall, mutational burden did not correspond to clinical response or to global demethylation kinetics (Fig. 7b, lower panels).

Finally, we asked whether there was any relationship between shift in specific mutated alleles and global demethylation across the corresponding treatment phase, focussing on the most highly mutated genes. We did not observe any correlation between demethylation and shifts in mutated allele frequency (Fig. 7c); furthermore, except for *TET2* (and possibly *TP53*), change in mutation frequency did not differ based on response status (Supplementary Fig. 10c). Thus, although sub-clonal structures of gene mutations showed temporal variation, most variant alleles persisted through both phases of treatment irrespective of clinical response or demethylation kinetics.

## Discussion
Here, we measured longitudinal in vivo drug incorporation and DNA methylation dynamics in a cohort of patients treated with injected and oral formulations of AZA. Drug incorporation was greater during the injection phase but was not directly proportional to DNA demethylation, which was greater and more sustained during the

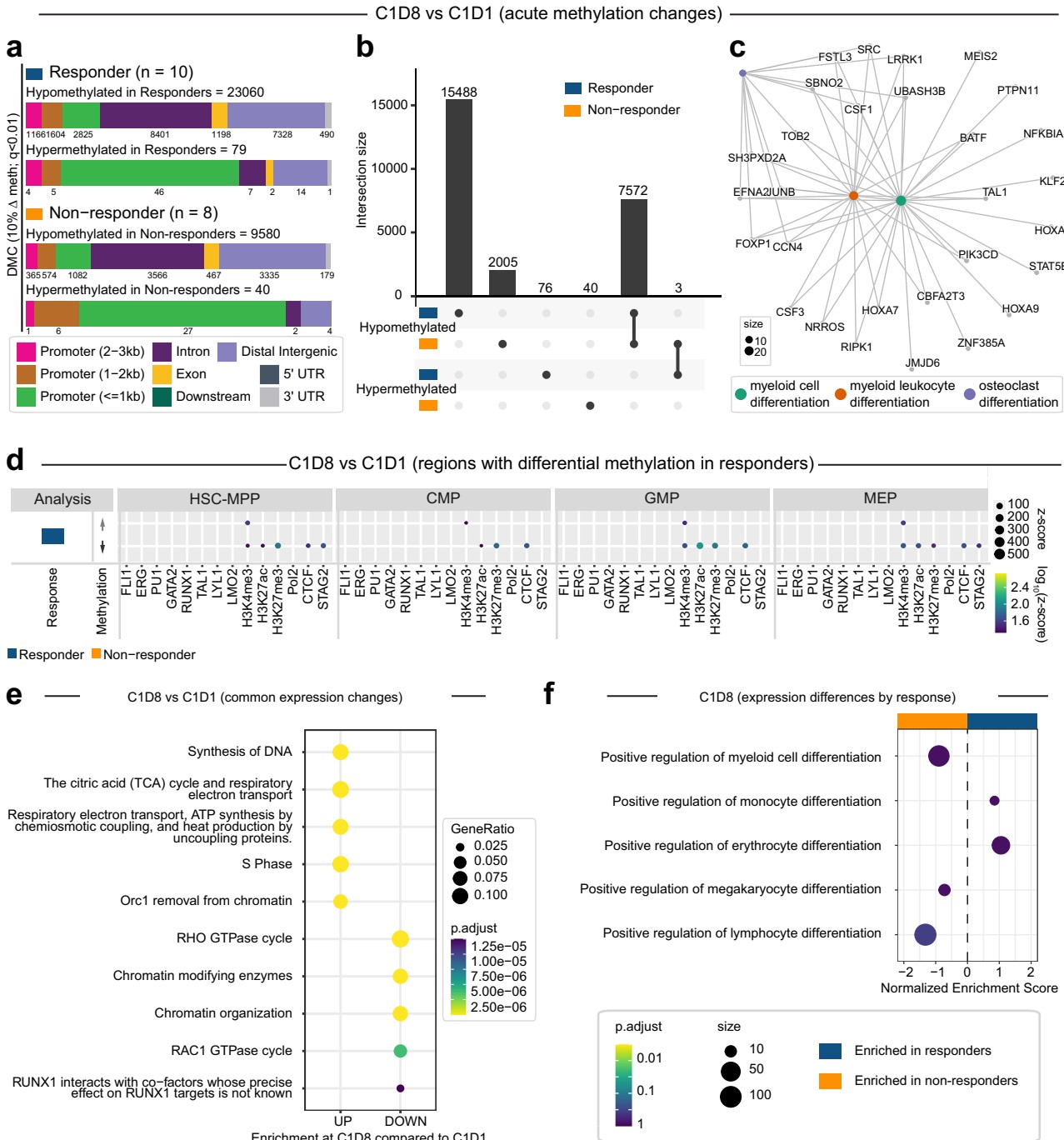

**Fig. 5 | Acute and sustained changes in site-specific CpG methylation in CD34+ HSPCs following AZA treatment. a–d** Acute methylation changes at 440140 CpG sites detected in all samples at C1D1 and C1D8 following the first cycle of AZA treatment. **a** Genomic distribution of CpG sites differentially methylated at C1D8 compared to baseline C1D1 in responders and non-responders. **b** Upset plot showing overlap of differentially methylated CpG sites between comparison groups. **c** Network diagram showing enriched pathways for genes mapping to CpGs which are uniquely hypomethylated in responders at C1D8 compared to C1D1 (n = 15488 CpGs). Network diagrams were created with clusterProfiler[91] using GO pathways[98]. CpGs were annotated to genes using HiChIP data from healthy human HSPC subsets[55]. **d** Enrichment of CpGs differentially methylated in responders at genomic regions with specific histone marks (H3K27Ac, H3K3me3, H3K27me3) or bound by key transcription factors and genome organisers (FLI1, ERG, PU.1, GATA2, RUNX1, TAL1, LYL1, LMO2, Pol2, CTCF, STAG2) in healthy human HSPC subsets. Plot shows CpG regions hypermethylated (upper) or hypomethylated (lower) at C1D8 compared to C1D1 in responders. **e, f** Changes in gene expression during cycle 1. **e** clusterProfiler[91] pathway[98] enrichment across all patients, irrespective of clinical response, at C1D8 vs C1D1. **f** Normalised enrichment scores from FSGEA[99] analysis for pathways related to blood production in responders compared to non-responders at C1D8.

oral phase. In this study, all patients were treated first with injected and then oral AZA, so we cannot rule out the possibility that prior treatment with injection AZA modulated incorporation and demethylation kinetics during the oral phase. Global DNA methylation levels in PB mononuclear cells were comparable in AZA responders and non-responders during their course of treatment. However, in responders we observed distinct baseline and early drug-induced methylation differences in CD34+ HSPCs that overlapped regulatory regions of genes associated with tissue patterning, cell migration, and myeloid differentiation. Following six treatment cycles,

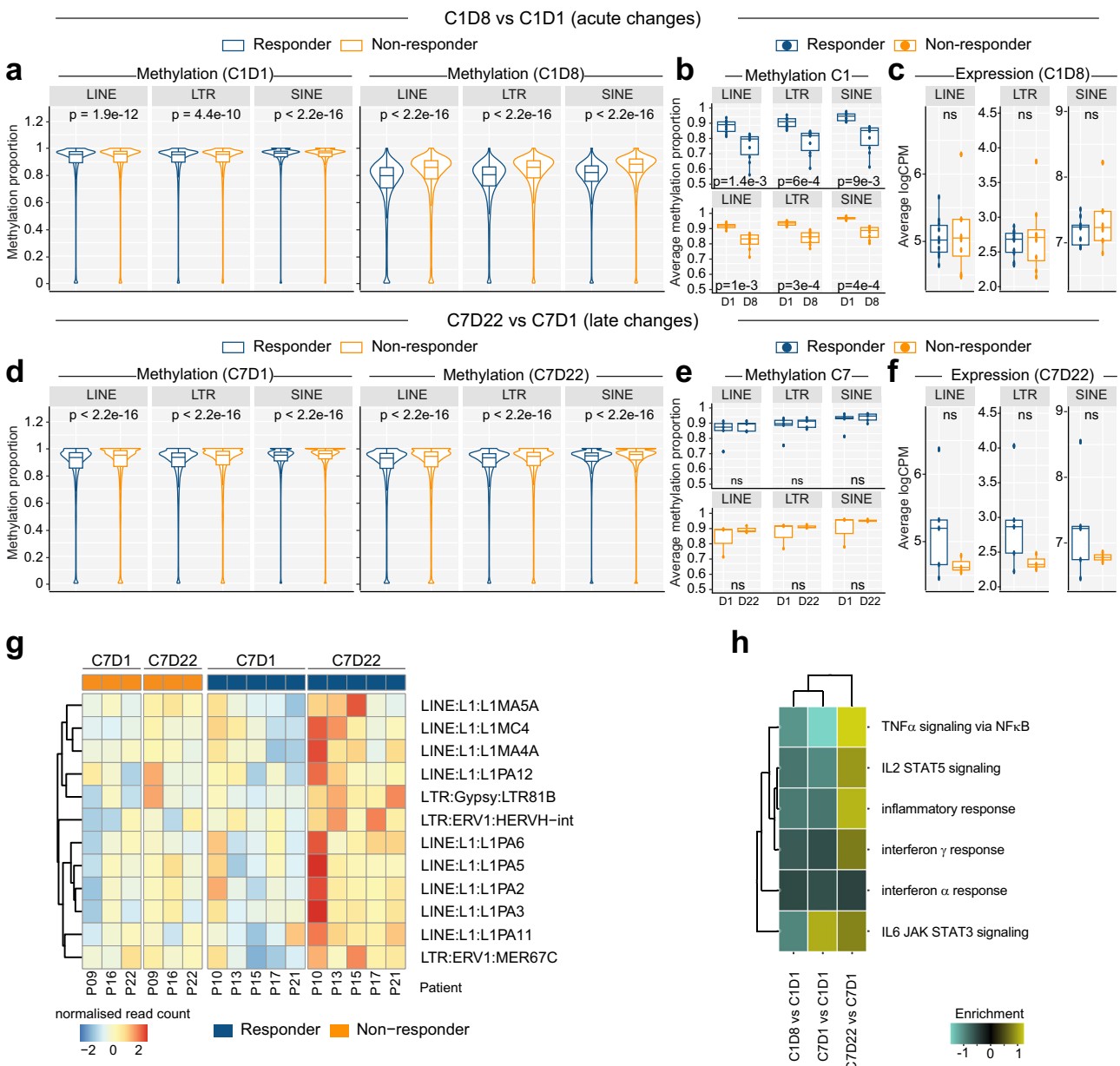

**Fig. 6 | Methylation and Expression changes at Transposable Elements (TE) in CD34+ HSPCs following AZA treatment. a–c** $n = 18$, responder: 10, non-responder: 8. **a** Methylation proportion (beta value) of CpGs mapping to TEs belonging to LINE, LTR, or SINE families in responders and non-responders. Left: at C1D1. Right: At C1D8. *P* values are indicated, statistical comparison used is two-sided Wilcoxon Rank-Sum test. Boxplots in (**a**, **d**) show centre = median, box = interquartile range (IQR), whiskers = furthest point within 1.5xIQR, outliers = not shown. **b** Average methylation proportion per patient in responders and non-responders at C1D1 and C1D8. *P* values are indicated, statistical comparison used is two-sided Wilcoxon Rank-Sum test. Boxplots in (**b**, **c**, **e**, **f**) show centre = median, box = interquartile range (IQR), whiskers = furthest point within 1.5xIQR, outliers = points >1.5xIQR. **c** Expression (average logCPM) for TEs belonging to LINE, LTR, or SINE families in responders and non-responders after one cycle of injected AZA (C1D8), statistics refer to two-sided student t-test ns indicates *P* > 0.05. **d–f** $n = 8$ patients, responder:

5, non-responder: 3. **d** Methylation proportion (beta value) of CpGs mapping to TEs belonging to LINE, LTR, or SINE families in responders and non-responders after one cycle of oral AZA. Left: at C7D1. Right: At C7D22. Statistical comparison used is two-sided Wilcoxon Rank-Sum test. **e** Average methylation proportion per patient in responders and non-responders at C7D1 and C7D22. Statistical comparison used is two-sided Wilcoxon Rank-Sum test, ns indicates *P* > 0.05. **f** Expression (average logCPM) for TEs belonging to LINE, LTR, or SINE families in responders and non-responders after one cycle of oral AZA (C7D22), statistics refer to two-sided student *t* test, ns indicates *P* > 0.05. **g** Per-patient expression of 12 TEs differentially expressed at C7D22 compared to C7D1. Differentially expressed TEs included evolutionarily young (PA1-PA6)[71,116] members of the L1PA# subfamily. **h** Summed enrichment scores for six hallmark inflammatory pathways in responders compared to non-responders.

differential hypomethylation in HSPCs mapped to genes involved in positive regulation of hemopoiesis and cell-cell adhesion. Contrary to previous reports[16], baseline cell cycle features did not predict patient response, although drug incorporation in the first treatment cycle was proportional to the number of cycling HPCs, and in responders, a greater proportion of HPCs exited quiescence and

continued to cycle at C12. Mutated alleles persisted across treatment cycles irrespective of clinical response, and temporal variations were not directly related to changes in DNA methylation.

Global DNA demethylation in PB and BM MNCs was not proportionate to in vivo DAC incorporation and was not a reliable measure of clinical response. By contrast, HSPCs in responders had lower baseline

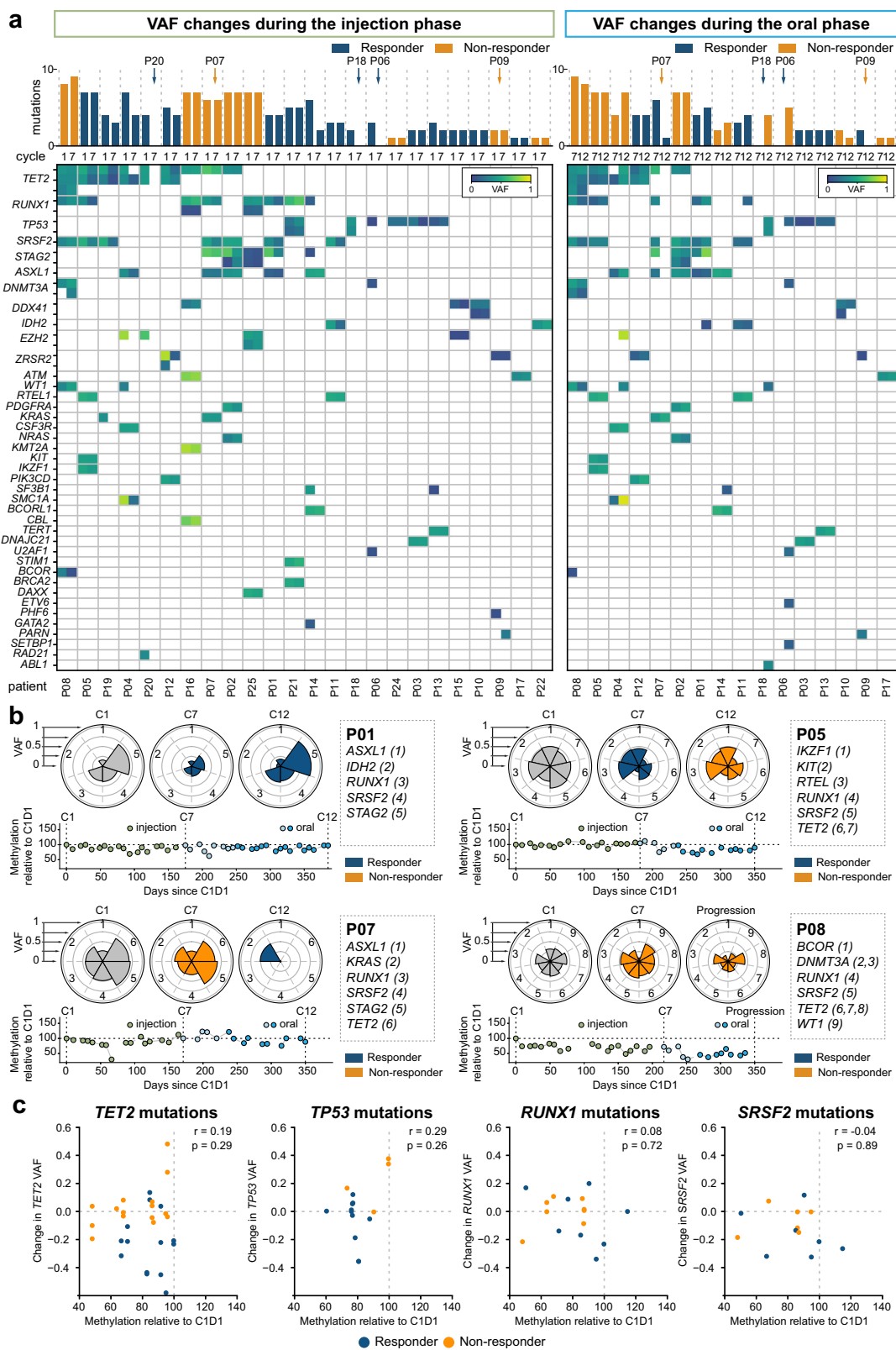

CpG methylation than in non-responders. Whilst a larger study is required to ascertain whether baseline CpG methylation at specific sites in HSPCs is predictive of clinical response, these data suggest that HSPCs in some individuals may be epigenetically primed to respond to AZA and point to pathways that may be associated with such a response. Indeed, hypomethylated DMCs in responders mapped to genomic regions with bivalent histone marks or bound by key

hematopoietic TFs in healthy HSPCs[55], features that are associated with drivers of myeloid and megakaryocyte-erythroid differentiation. The origin of baseline CpG methylation differences in HSPCs, and whether they are related to specific genetic or epigenetic features, or the phylogenetic age of the mutated clone, is unclear. However, these data provide a foundation for more granular investigations of DNA methylation and sub clonal heterogeneity at single cell resolution[68].

**Fig. 7 | Variations in clonal composition in BM MNCs during HMA treatment.**
**a.** Variant allele shifts in each patient over (left) injection phase, and (right) oral phase. Each column shows an individual patient with heatmap representation of VAF for each variant allele at start and end of the treatment phase (or, where relevant, during the oral phase, at progression). Bar plots show total number of variant alleles detected at each time point and are coloured by clinical response across the relevant treatment phase. **b** Clonal shifts in representative patients. Upper panels: Nightingale plots show VAF for each variant at diagnosis (C1), following 6 treatment cycles (C7) and following 12 treatment cycles/at progression (C12). Lower panels: Methylation level relative to C1D1 in PB across the entire course of treatment. **c** Correlation between global changes in methylation and shifts in variant allele burden for the four most frequently mutated genes in this cohort. Plots show composite data; each data point indicates a change in VAF across either injection or oral treatment phase and the average global methylation relative to C1D1 over the same treatment phase. r = two-sided spearman correlation coefficient.

In healthy HSPCs, key TFs are pre-emptively bound at regulatory regions of lineage-specific genes expressed at later developmental timepoints[55]. CpG hypomethylation in MDS HSPCs during the first AZA cycle was skewed to non-coding regions, and in responders, mapped to pathways involved in activating myeloid programmes. Methylation changes at these regions may have altered the gene regulatory landscape, facilitating myeloid differentiation and improved circulating blood counts. Following multiple cycles of AZA when response was clinically apparent, methylation differences between response groups persisted, although only a small set of CpGs were hypomethylated compared to baseline. These mapped to pathways including cell-cell adhesion and positive regulation of hemopoiesis and leucocyte differentiation that pointed to altered relationships with other cells in the bone marrow microenvironment. Importantly, in responders, we also observed increased expression of gene pathways associated with myeloid and erythroid differentiation. Gene expression changes at late time points also included increased expression of evolutionarily young members of the PA family of LINE TEs. TEs may influence high level genome organisation[69] and erythropoiesis[70]. Furthermore, members of the PA family have been implicated in long range gene expression and induction of a type I interferon response[71–73], and activation of young TEs has been linked to clinical response to AZA[74]. Accordingly, inflammation-related pathways were also upregulated in responders. Overall, the observed alterations in locus specific CpG methylation and gene expression during AZA treatment are consistent with biological processes underlying clinical response.

Incorporation of HMAs into DNA is S phase dependent[15], and cell cycle features in HSPCs have been linked to patient response[16,47,75–77]. Although baseline cell cycle state in HSPCs did not predict patient response in this cohort, a significantly greater proportion of cells exited quiescence (C7 vs C1) during treatment and continued to cycle at C12 in AZA responders versus non-responders, underlining the continued need for S phase dependent drug incorporation for response. It is unclear whether these changes are a direct response to HMAs or whether they reflect the increased progenitor proliferation underlying clinical response, but any drug combination that abrogates S phase dependent incorporation of HMAs will likely dampen their clinical efficacy.

MDS/CMML are driven by somatic mutations in HSCs, and there is increasing evidence that highly mutated stem cells with altered differentiation capacity contribute to improved peripheral counts in responders[42]. Indeed, most patients in this cohort had stable clonal composition over treatment phases, regardless of response status, drug incorporation, or global demethylation. What remains unclear is whether specific CpG demethylation and improved differentiation capacity occur in most cells or whether reprogramming of a small fraction of mutant stem cells is sufficient to improve circulating output. This has implications for disease progression as mutated stem cells that do not regain differentiation capacity may persist as a cellular reservoir for progressive disease.

Overall, this study revealed that clinical response was not related to how much global demethylation occurred during treatment, but rather to demethylation of specific CpG sites in HSPCs. Due to limitations with sample material, we only evaluated differences in pooled CD34+ fractions using RRBS; genome-wide evaluation of CpG methylation in HSPC subfractions may reveal further insights into differentially methylated CpGs that regulate HMA response. Since CpG methylation varies with cell lineage and maturation[78], differences in the CD34+ progenitor composition may well impact interpretation of the current data. Furthermore, this study focussed on cell-intrinsic effects in HSPCs and their progeny. HMAs potentially affect any replicating cell in the BM, and the interplay between the BM microenvironment and mutated HSPCs are involved in pathogenesis of MDS[79–82]; thus, AZA-induced effects on additional cell types in the bone marrow may modulate clinical response to HMAs. Finally, the relatively small sample size of our cohort limits our ability to detect small differences, and in particular, the locus-specific methylation differences we observe should be validated in a larger study. Nevertheless, this study uncovered specific regions in HSPCs where demethylation was associated with therapeutic effects, and such regions are of value not only as a gauge for early HMA response assessment but also as potential sensors for screening novel agents or as targets for site-directed demethylation therapies.

## Methods

The research in this manuscript complies with all relevant ethical regulations. The trial protocol received ethical approval (Ref#: 17/295, 2019/ETH05009) from the South Eastern Sydney Local Health District Human Research Ethics Committee, and participating sites received Institution approval to conduct the trial prior to commencing recruitment.

### Trial design, patients, and oversight

Participants were recruited to an open label phase 2 multicentre investigator-initiated trial (NCT03493646: Evaluating in Vivo AZA Incorporation in Mononuclear Cells Following Vidaza or CC-486). Participants who commenced treatment received 6 cycles of parenteral AZA (75 mg/m[2]/day administered subcutaneously for 7 days followed by a rest period of 21 days) followed by 6 cycles of oral AZA (2 cycles at 100 mg bid, 4 cycles at 150 mg bid for 21 days, followed by a rest period of 7 days) (Fig. 1a), and were followed up for a further 12 months. Patients were eligible for enrolment following a new diagnosis of HR-MDS (IPSS; intermediate-2 or high-risk), CMML (bone marrow [BM] blasts 10–29%), or LB-AML (20–30% blasts). Extended eligibility criteria are available at https://clinicaltrials.gov/study/NCT03493646. 53 patients were assessed for eligibility and 40 commenced treatment (Fig. 1b). Pharmacodynamic sampling was performed throughout the treatment period (Fig. 1a), and clinical response assessed at the end of cycle 6 (C6D28) and end of cycle 12 (C12D28). Clinical response was assessed by treating clinicians using IWG2006 guidelines[50] and independently verified by trial monitors and adjudicated by a panel where there was disagreement. For analysis purposes, patients were classified by their response in the corresponding phase. For example, P10 was grouped with responders for the injection phase and non-responders for the oral phase (Fig. 1c). Response status during the oral phase is based on IWG assessment at end of cycle 12 when available (n = 11); patients who progressed during the oral phase or soon after stopping treatment were assessed as non-responders (n = 6), and remaining patients were assigned response as per end of cycle 6 IWG assessment (n = 2). Likewise, patients who progressed during the injection phase and prior to IWG assessment were considered non-responders. No sub-

analysis was performed by sex as the study design was not sufficiently powered.

When indicated, best supportive care was allowed in combination with study treatment and included red blood cell or platelet transfusion, antibiotics, antifungals, antivirals, anti-emetics, anti-diarrhoeal agents, and granulocyte colony-stimulating factor (G-CSF). Use of erythropoiesis stimulating agents and hematopoietic growth factors other than G-CSF was excluded. Patients were monitored prior to commencement of each treatment cycle with a physical exam, vital signs, haematology, biochemistry, liver function, and ECOG assessment. Criteria for discontinuation included lack of efficacy, progressive disease, withdrawal by participant, adverse event(s), death, lost follow-up, protocol violation, study termination by the sponsor, pregnancy, recovery, non-compliance with azacitidine, transition to commercially available treatment, physician decision, disease relapse, and symptomatic deterioration. Actual reasons for discontinuation were as shown (Fig. 1b).

The trial was funded by Celgene and sponsored by the University of New South Wales. The Kirby Institute, UNSW was responsible for trial coordination. The protocol received ethical approval (Ref#: 17/295, 2019/ETH05009) from the South Eastern Sydney Local Health District Human Research Ethics Committee, and participating sites received Institution approval to conduct the trial prior to commencing recruitment. All participants provided written informed consent for clinical research including genetic testing and collection/storage of human tissue. A protocol steering committee oversaw the trial, and independent adjudicators confirmed clinical response assessments provided by attending clinicians. Serious adverse events (SAE) were recorded during all cycles (Supplementary Tables 6 and 7). SAEs reported during the oral AZA cycles were assessed by independent medical monitors. J.O. led the statistical analyses with assistance from J.A.I.T., F.Y., and F.C.K. All authors had access to trial data.

### Primary and secondary trial outcomes

The primary objective was to determine whether there is greater AZA incorporation in DNA following 21 days of oral AZA compared to 7 days of injection AZA in a 28-day treatment cycle, and whether incorporation was associated with greater clinical and/or molecular response. Power calculations indicated 90% power to detect a standardised difference of 0.77 with $n = 20$.

Secondary objectives of the trial were based on a previous small study where patients who responded to injection AZA had a greater fraction of cycling HPCs compared to patients who fail to respond[16]. However, whether increased replication is associated with increased AZA incorporation was not known. The availability of an assay[47] to measure AZA incorporation and the ability to measure the fraction of replicative HPCs forms the basis of the secondary objectives of this study.

Raw data for primary and secondary outcomes was acquired by individuals blinded to patient outcome.

### Statistical analyses

Statistical comparisons for cell cycle, AZA-MS in BM, LINE-1 qPCR, and VAF data were performed in Python (v3.8.3) using the scipy (v1.10.1) stats module. Single timepoint comparisons between outcome groups used the function stats.ttest_ind() using default parameters (ie/ two-sided tests), or where stated, using equal_var = False (Welch's ttest, two-sided). Correlations were tested using the function stats.-spearmanr() using default parameters (ie/two-sided tests). $P$ values < 0.05 were considered significant. Boxplots were generated using default sns.boxplot parameters (centre line = median, box limits = interquartile range (IQR), whiskers = the most distant data point within 1.5× IQR, outliers = data points >1.5× IQR).

Drug incorporation into DNA and global DNA methylation levels were compared for responders and non-responders, or for injected

and oral phases, or between cell-cycle timepoints using a linear mixed model approach (two-sided) with subject-level random effects to account for within-subject variability. Drug incorporation into PB cells and global DNA methylation levels were compared over collection timepoints (D1, D8, D22) for responders and non-responders, or for injection and oral phases, using a linear mixed model approach (two-sided) with subject-level random effects to account for within-subject variability. Oral phase analysis of drug incorporation and DNA methylation was restricted to cycles 9 and later to minimise the likelihood of detecting changes carried over from the injection phases. Cell cycle parameters across timepoints, within and between responder groups, were compared using two-sided linear mixed models using random intercepts by patients to account for within-subject variability A gamma test was conducted to assess whether the proportion of responders increases or decreases as BMT cellularity increases, and a two-sided linear model was fit to %CD34+ HSPC by responders (y vs n), cellularity and their interaction term to assess any relationship between BMT cellularity and %CD34+ HSPC in BMA. All these statistical approaches utilised SAS (9.4).

### Processing of BM and PB

Peripheral blood (20 mL per timepoint in addition to standard of care collection) was collected in 10 mL EDTA vacutainers. Immediately following collection, 1 mL of 1 mg/mL tetrahydrouridine (THU; abcam #ab142080) was added per tube, and the specimen rocked at 4 °C until processing. Blood was centrifuged at $800 \times g$ for 10 min at room temperature, upper plasma layer removed and stored at −80 °C, and the remaining red cell/buffy coat fraction treated with two rounds of 1X red blood cell lysis buffer (BD Pharm Lyse #555899). Mononuclear cells (MNCs) were recovered by centrifugation at $800 \times g$ and pellets snap frozen at −80 °C for subsequent DNA and RNA isolation.

Bone marrow aspirates (BMA; 30 mL per timepoint in addition to standard of care collection) were collected into 10 mL Hanks balanced salt solution (Life Technologies #14170-161) containing 100IU/mL porcine heparin (Pfizer). Immediately following collection, 4 mL of 1 mg/mL THU was added, and the specimen rocked at 4 °C until processing. BMA samples were diluted 1:4 with RPMI medium (Gibco #21870092), underlaid with 10 mL Lymphoprep (Axis-Shield #1114547), and centrifuged at $800 \times g$ for 35 min at RT. The upper plasma layer was removed and snap frozen at −80 °C, and MNCs collected for storage (pellets for DNA/RNA isolation or viably frozen) or further processing. CD34+ cells were isolated using CD34+ microbeads (Miltenyi #130-046-703) and an AutoMACS machine (Miltenyi). Cell purities were monitored using flow cytometry (APC-CD34 Miltenyi # 130-113-176) and CD34+ and CD34 depleted cells were viably frozen for later use.

### Isolation of DNA and RNA

DNA and RNA were isolated from frozen cell pellets or thawed CD34+ cells using either an All-in-One DNA/RNA miniprep kit (BioBasic #BS88203) for mass spectrometry experiments or an AllPrep DNA/ RNA mini kit (Qiagen #80204) for RRBS and RNAseq experiments. DNA was quantified using either a Nanodrop spectrophotometer (Thermo) for mass spectrometry DNA samples and RNA samples or a Qubit fluorometer (Thermo) using a Quant iT Qubit dsDNA HS Assay Kit (Invitrogen #Q32851) for RRBS DNA samples.

### Cell cycle analysis

Up to two million thawed bone marrow mononuclear cells (MNCs) were washed and resuspended in flow cytometry buffer (2% foetal bovine serum, 1 mM EDTA in PBS) containing human Fc block (BD Biosciences). Cells were first stained with a PE-Cy5 lineage marker cocktail (CD2 (clone RPA-2.10), CD3 (clone HIT3a), CD10 (clone HI10a), CD19 (clone HIB19), CD20 (cone 2H7) and GPA/CD235ab (clone HIR2); BioLegend) for 15 min on ice, then washed and fixed with 1.6%

paraformaldehyde for 10 min at room temperature and permeabilised using 90% ice-cold methanol for 30 min on ice. Fixed, permeabilised cells were then incubated with CD34-PE (clone 8G12, BD Biosciences), CD38-APC-Cy7 (clone HIT2, BioLegend), and Ki67-AF488 (clone B56, BD Biosciences) antibodies for 1 h at room temperature in the dark. Cells were washed twice, then stained in 0.5 µg/mL of 4′,6-diamidine-2-phenylidole dihydrochloride (DAPI, BD Biosciences) for 15 min at room temperature and data acquired on a LSRFortessa (BD) flow cytometer at the Mark Wainwright Analytical Centre (UNSW Sydney). Data was analysed using FlowJo software (Tree Star, USA). HSC samples containing less than 50 cells were excluded from analysis.

## Measurement of DAC incorporation and DNA methylation using mass spectrometry (AZA-MS)

DNA DAC incorporation and demethylation was measured by mass spectrometry (AZA-MS)[47] at all collection time points. Briefly, 250 pmol 5-Aza-2′-deoxy Cytidine-15N4 (15N-DAC; Toronto Research Chemicals # A79695) was spiked-in to 5–10 µg purified DNA in a total volume of 30 µL. After addition of 10 µL of 20 mg/mL NaBH₄, samples were incubated for 20 min at room temperature with agitation. The pH was then neutralised by addition of 1 µL 2 M HCl. DNA was then fragmented by addition of 30 µL of fragmentation mixture (100 U/mL benzonase (Sigma-Aldrich #E1014), 240 U/mL phosphodiesterase I (Sigma-Aldrich #P3243), 40 U/mL alkaline phosphatase (Sigma-Aldrich #P7923), 20 mM Tris-HCl pH 8.0, 20 mM NaCl, 3 mM MgCl₂) and incubation for 1 h at 37 °C. Fragmented samples were dried under vacuum (Savant Speedvac Plus SC210A, Thermo) and stored at −80 °C prior to mass spectrometry analysis. Standard curves for all measured analytes (Decitabine (DAC; Selleck Chemicals #S1200), Cytidine (C; Selleck Chemicals #S2053), 5-methylcytidine (mC; Cayman Chemical #16111), 2′deoxycytidine (dC; Cayman Chemical #34708), 2′deoxy-5-methyl-cytidine (mdC; MMP Biomedicals # 0219888310-10 mg)) were prepared in parallel and covered the following ranges; DAC, 0.3–40 pmol, C/dC, 3–100 pmol, mC/mdC, 0.06–2 pmol, all per 20 µL injection.

Samples were reconstituted in 50 µL CE buffer (10 mM Tris HCl pH 9.0, 0.5 mM EDTA) and run on a Q Exactive PLUS Orbitrap mass spectrometry system (Thermo) using a heated electrospray interface operated in the positive ion mode. Chromatographic separation was performed on a 100 mm × 2.1 mm i.d., 3 µM, C30 column (Acclaim, Thermo) at 40 °C. Duplicate 20 µL injections of each sample were analysed using gradient elution with 0.1% formic acid in Milli-Q water (Solvent A) and 0.1% formic acid in acetonitrile (Solvent B) at a flow of 0.4 ml/min over 8 min. Mass spectra were acquired at a resolution of 140,000 over the range of 220 to 260 Th with the electrospray voltage at 4000 V. Sheath gas pressure and auxillary gas pressure were 27 and 10. The capillary temperature was 300 °C and the s-lens was 80 V. Data processing of chromatograms was performed using the Quanbrowser function of the Xcalibur Software package version 2.5 (Thermo) and analyte measurements used to calculate pmol DAC detected per µg of input DNA and DNA methylation relative to C1D1 levels ([mdC/dC timepoint] ÷ [mdC/dC C1D1]).

## qPCR-based measurement of DNA demethylation at LINE-1 elements

LINE-1 qPCR was performed essentially as described[53] and using the same primers except for a modified HpaII-cut specific reverse F-oligo: TGGCTGTGGGTGGTGGGCCTCGTAGAGGCCCTTTTTTGGTCGGTACC TCAGATGGAAATGTCTT/3ddC/. Technical replicates of up to 4 ng of DNA in PCR master mix (20 mM Tris-HCl pH 8.4, 50 mM KCl, 7 mM MgCl₂, 0.2 mM dNTP mix (dATP, dCTP, dGTP, dTTP), DraI-cut specific oligonucleotides (12.5 nM each of forward and reverse F-oligos plus 160 nM each of forward and reverse outer primers and HEX probe), HpaII-cut specific oligonucleotides (2 nM each of forward and reverse F-oligos plus 40 nM each of forward and reverse outer primers and FAM probe), 2U HpaII (NEB #R0171L), 1U DraI (NEB #R0129L), 0.5U Hot Start Taq DNA Polymerase (NEB #M0495L)) were cycled at 37 °C for 15 min; 90 °C for 5 s; 95 °C for 2 min; then 10 cycles of 90 °C for 5 s, 95 °C for 15 s, 60 °C for 1 min, and 68 °C for 20 s; followed by 40 cycles of 95 °C for 15 s, 65 °C for 40 s, and 68 °C for 20 s in a CFX96 real-time PCR machine (BioRad). Efficiency of the FAM and HEX reactions was calculated using a standard curve over the range 6.25 pg–4 ng per reaction, generated from AZA-treated cell-line DNA. The LINE-1 demethylation level for each cycle of each patient was normalized to the corresponding C1D1 sample using the following formula: Relative LINE1 demethylation = (Efficiency (FAM)$^{Ct(FAM, C1D1)-Ct(FAM)}$)/(Efficiency (HEX)$^{Ct(HEX, C1D1)-Ct(HEX)}$).

## Reduced representation bisulphite sequencing (RRBS)

RRBS libraries were constructed from 50 ng genomic DNA using an Ovation RRBS Methyl-Seq system (Tecan). Bisulphite conversion was performed using the Epitect Fast DNA Bisulfite kit (Qiagen) and libraries amplified using the Ovation RRBS Methyl-Seq system (Tecan). Libraries were sequenced on a Novaseq 6000 using the SP100 flow cell (Illumina) at Monash University's School of Translational Medicine Genomics Facility.

Raw RRBS data in fastq format were quality and adapter trimmed according to the manufacturer's guide using trim_galore (0.6.4) and a custom trimming script[83]. The trimmed fastq files were then aligned to a bisulfite converted genome (hg38) using Bismark (0.22.3) and methylation status at each CpG locus were extracted[84]. Differentially methylated cytosines (DMCs) were identified using methylKit (1.22.0), incorporating patient ID as a covariate when available and differentially methylated regions (DMRs) were identified with edmr (0.6.4.1) packages in R (4.2.0)[85,86].

DMCs and DMRs were annotated using ChIPseeker (1.32.1) for the nearest gene and annotatr (1.22.0) for CpG island annotation[87,88]. DMCs were further annotated using HiChIP data from healthy HSPC subsets[55]. DMCs were first linked to significant HiChIP interaction at 5 kb resolution, and then associated with genes if the DMC-containing bin was within 10 kb of the TSS of specific genes. Gene ontology[89,90] enrichment was performed using HiChIP associated genes, and visualised using clusterProfiler (4.11.0.001)[91]. ChIP-seq peaks[55], including TF and histone modifications from different cell types, were used to evaluate the enrichment of DMCs in key regulatory regions. DMCs were first expanded 100 bp on both sides and overlapped regions were merged. The regioneR package (1.28.0) was used to generate 100 sets of randomised regions with matched characteristics, and the z-score was calculated as (observe − mean)/sd and visualised[92]. Bigwig files were generated using Bedtools[93]. Composite RRBS tracks are available for visualisation using the UCSC browser at the following link https://genome.ucsc.edu/s/PimandaLab/OralAZA_RRBS.

## RNA sequencing

Extracted RNA was amplified using a SMARTer universal low input RNA kit (Takara) and used to prepare standard poly A selected sequencing libraries (Novogene, Singapore). Libraries were sequenced using a NovaSeq600 (Illumina) by Novogene. Following QC assessment with fastqc (0.11.9), RNAseq adapters were trimmed using picard (2.26.2), reads aligned to the hg38 genome using the STAR aligner[94] (2.7.9), and the count matrix generated using featureCounts from subread[95] (2.0.2). Differential gene expression was determined using edgeR[96] (4.3.0) in R (4.4.0); genes with absolute log fold change>2 and P < 0.01 (unadjusted) were considered differentially expressed. Barcode plots were generated using limma[97] (3.61.9) and the indicated gene sets[16,52]. Pathway enrichment was performed and visualised using clusterProfiler (4.11.0.001)[91] and gene sets from the Reactome Pathway Database[98]. FGSEA (1.30.0) analysis[99] was performed using selected gene ontology[89,90] pathways (GO:0045639 (positive regulation of myeloid cell differentiation), GO:0045657 (positive regulation of monocyte differentiation), GO:0045648 (positive regulation of

erythrocyte differentiation), GO:0045654 (positive regulation of megakaryocyte differentiation), GO:0045621 (positive regulation of lymphocyte differentiation)). Pathways associated with specific gene clusters at C1D1 were determined using Metascape[100].

FASTQ files were quantified with Salmon (v 1.10.2)[101] using Gencode v44 comprehensive gene annotation. We then used edgeR::catchSalmon (v 4.2.0) to import counts and read to transcript ambiguity[96]. The scaled transcript counts (counts divided by read-to-transcript ambiguity) were used for differential transcript analysis (DTE and DTU)[102]. We filtered out lowly expressed transcripts and used edgeR::glmQLFit for DTE and edgeR::diffSpliceDGE for DTU analysis. For DTU analysis, gene-level $P$ values were obtained with Simes correction. We used FDR cutoff 0.2 to select significant transcripts or genes.

To perform differential splicing analysis, we first used STAR (v 2.7.11b)[94] to map FASTQ files to GRCh38 reference genome in two pass mode. Then BAM files were input into rMATS (v 4.3.0)[103,104] with Gencode v44 comprehensive gene annotation to obtain differential splicing events. Paired test mode was used when evaluating same patients across timepoints. We used FDR cutoff 0.05 to obtain significant events. Gene ontology analysis was performed using clusterProfiler[91].

TE locations were identified using RepeatMasker[105] (4.1.5) and used by the featureCounts function of R subread[95] (2.8.2) to give read counts at all annotated TE regions. Read counts for individual TEs were pooled into Families and Subfamilies. Differential Expression was carried out on the combined TE families/subfamilies and gene read counts using DESeq2[106] (1.34.0), weighted using Independent Hypothesis Weighting (IHW)[107]. Differentially expressed TE families and subfamilies were identified based on absolute log2foldchange > 0.4 and P ≤ 0.05 and a maximum RPM ≥ 100. For pair-wise comparisons across different time points/treatment cycles, ranked lists of genes based on the log2foldchange computed in the DESeq2 analysis were used for gene set enrichment analysis (GSEA)[108] (4.3.0) using the MSigDB[108–110] HALLMARK gene set.

## Myeloid capture panel sequencing and analysis

DNA from BM MNCs at baseline and clinical response timepoints was utilised for assessment of VAFs using a myeloid gene capture panel with sensitivity to detect VAFs at or above 5%. Capture sequencing library construction and sequencing was performed at the University of Auckland using standard clinical myeloid capture arrays (Table S10). Sequencing data quality was assessed with fastqc tool (v0.11.5) followed by BWA alignment to the hg19 reference genome (v0.7.12). Sam files were converted to bam format and sorted using samtools (v1.3.1) and mpileup files generated using the following parameters: maximum depth (-d) of 1000, minimum base quality (-Q) of 15 and minimum mapping quality (-q) of 10. Variants were called using Varscan (v 2.3.9) to generate VCF (variant call format) files using the following parameters: minimum coverage (--min) of 8, minimum depth of the reads supporting the variant (--min-reads2) of 2, minimum variant frequency (--min-var-freq) of 0.01 and strand filter (--strand-filter) 0. Variants were annotated using ANNOVAR[111] and SnpEffect[112].

Variants were filtered to remove SNP138 non-flagged variants except for rs377577594 (corresponding to *DNMT3A* R882). Furthering filtering was performed to remove variants occurring at a minor allele frequency of greater than 0.1% in gnomAD[113], ExAC[114], and 1000 genomes[115] databases. Then any variant belonging to a blacklist of known false positives and any variant at a VAF of less than 5% were excluded.

## Data availability

The RRBS and RNA sequencing data generated in this study are publicly available and have been deposited in the GEO database under accession codes GSE274997 and GSE274999. Relevant clinical data (bone marrow features, cell counts, and additional response information) are available in the Supplementary Tables file. Additional patient demographic information plus flow cytometry, nucleotide mass spectrometry, PCR, and variant allele frequency data used to generate the conclusions reported in this study are available in the Source Data file. Source data are provided with this paper.

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

## Acknowledgements

Patients and their carers, clinicians, and trial coordinators at recruiting sites, Celgene/BMS (Dr Jessica Morison, Dr Tamara Etto, Ms Dharini Fernando, Dr Kyle MacBeth, Dr Kevin Lynch, Dr Han Mynt, Dr Du Lam, Ms Tracey Crivellaro, and Dr Amir Samuel), UNSW (Mr Alan Melrose and Dr Qiao Qiao), SESLHD (Ms. Deborah Adrian and Ms Karen Mccardie), and Scientia Clinical Research (Ms Lisa Nelson) for input during protocol development, trial set up and follow-up. Some of the data presented in this work was acquired by personnel and/or instruments at the Mark Wainwright Analytical Centre (MWAC) of UNSW Sydney, which is in part funded by the Research Infrastructure Programme of UNSW. The investigator initiated clinical trial was funded in part by Celgene/BMS (RG172029 to J.E.P) with research support from the National Health and Medical Research Council (RG170246, RG211412 to J.E.P.), Anthony Rothe Memorial Trust (RG182042, RG202657, RG213236 to J.E.P, J.A.I.T), Leukaemia Foundation (RG231257 to J.E.P.). Patients were eligible for and received injection azacitidine from the pharmaceuticals benefit scheme (Australia) and CC-486 from Celgene/BMS.

## Author contributions

J.E.P designed the research study and served as the lead PI on the clinical trial. C.L. wrote the trial protocol and M.N.P. oversaw the clinical trial. J.A.I.T., F.Y., H.R.H., S.J., J.S., C.H.S., X.Y.L., A.C.N., P.M.K., G.S.B., X.Z., M.N., E.S.G., F.C.K. and I.R.T. performed research and analysed data. J.A.I.T., F.V., M.M.K., R.P., M.J.R., C.J.J., S.K.B., D.J.C., J.W.H.W., A.U., M.N.P., and J.E.P. provided research supervision. S.D. and S.H. coordinated the clinical trial. G.B., M.B., Y.G., N.A. assisted with trial coordination and database design. L.V., L.P., C.Y.F., M.K., D.K.H., R.I.S., S.M., L.L., F.H.P., S.R.L., K.J.S., K.K.S., P.R., P.P., W.S.S., S.L., C.T., S.J.F., F.R., A.K.E., D.H. and M.H. assisted with the trial. J.O. performed and oversaw statistical analysis. J.A.I.T and J.E.P. wrote the manuscript. All authors have viewed and approved the manuscript.

## Competing interests

F.V. is affiliated with OmniOmics.AI Pty Ltd. C.F. is an advisory board member at Amgen, AbbVie, Adaptive Biotech, BeiGene, Pfizer, Otsuka, and Jazz, a consultant at Novotech, and received speaker fees from Amgen, Pfizer, Servier, BMS, and Astella. D.H. has consultancy agreements with GlaxoSmithKline and Pharming Corp. M.H. is a consultant/advisory board member at Roche, Gilead, Otsuka, Janssen, Beigene, and Takeda. M.N.P. received research funding and/or provision of drug for clinical trials (to institution) from AstraZeneca, BRII Biosciences, Celgene/BMS, CSL Behring, Eli Lilly, Emergent Biosciences, Gilead Pharmaceuticals, GlaxoSmithKline, Grifols, Janssen/Johnson and Johnson, Takeda, ViiV Pharmaceuticals and has advisory roles with Celgene/BMS, Gilead Pharmaceuticals, and ViiV Pharmaceuticals. J.E.P. received research funding and/or provision of drug for clinical trials (to institution) from Celgene/BMS, Astex, Verastem Oncology and received honoraria from Abbvie as an advisory board member. The remaining authors declare no competing interests.

## Additional information

**Julie A. I. Thoms** [ORCID] [1] [✉], **Feng Yan** [ORCID] [2], **Henry R. Hampton** [ORCID] [1], **Sarah Davidson** [3,4], **Swapna Joshi** [5], **Jesslyn Saw** [6], **Chowdhury H. Sarowar** [5], **Xin Ying Lim** [5], **Andrea C. Nunez** [5], **Purvi M. Kakadia** [7], **Golam Sarower Bhuyan** [5], **Xiaoheng Zou** [1], **Mary Nguyen** [1], **Elaheh S. Ghodousi** [1], **Forrest C. Koch** [8], **Fatemeh Vafaee** [ORCID] [8,9], **I. Richard Thompson** [ORCID] [10],

Mohammad M. Karimi [10], Russell Pickford[11], Mark J. Raftery[11], Sally Hough[4], Griselda Buckland[3,4], Michelle Bailey[4], Yuvaraj Ghodke [3,4], Noorul Absar[4], Lachlin Vaughan[5,12,13], Leonardo Pasalic[12,13], Chun Y. Fong[14], Melita Kenealy[15], Devendra K. Hiwase [16], Rohanna I. Stoddart[17], Soma Mohammed[13], Linda Lee[18], Freda H. Passam[19,20], Stephen R. Larsen[20], Kevin J. Spring [21,22], Kristen K. Skarratt [23], Patricia Rebeiro[24], Peter Presgrave[25], William S. Stevenson[18], Silvia Ling[26], Campbell Tiley[27], Stephen J. Fuller [23], Fernando Roncolato [28], Anoop K. Enjeti[29,30,31], Dirk Hoenemann[32], Charlotte Lemech[33], Christopher J. Jolly[1], Stefan K. Bohlander [7], David J. Curtis [6], Jason W. H. Wong[34], Ashwin Unnikrishnan [5], Mark Hertzberg[35], Jake Olivier [36], Mark N. Polizzotto[3,4] ✉ & John E. Pimanda [1,5,35] ✉

[1]School of Biomedical Sciences, University of New South Wales, Sydney, NSW, Australia. [2]Bioinformatics Division, Walter and Eliza Hall Institute of Medical Research, Parkville, VIC, Australia. [3]ANU Clinical Hub for Interventional Research (CHOIR), John Curtin School of Medical Research, Canberra, ACT, Australia. [4]Kirby Institute, University of New South Wales, Sydney, NSW, Australia. [5]School of Clinical Medicine, University of New South Wales, Sydney, NSW, Australia. [6]Australian Centre for Blood Diseases, Monash University, Melbourne, VIC, Australia. [7]Leukaemia & Blood Cancer Research Unit, Department of Molecular Medicine and Pathology, Faculty of Medical and Health Sciences, The University of Auckland, Auckland, New Zealand. [8]School of Biotechnology and Biomolecular Sciences, University of New South Wales, Sydney, NSW, Australia. [9]UNSW Data Science Hub, University of New South Wales, Sydney, NSW, Australia. [10]Comprehensive Cancer Centre, School of Cancer and Pharmaceutical Sciences, King's College London, London, UK. [11]Bioanalytical Mass Spectrometry Facility, University of New South Wales, Sydney, NSW, Australia. [12]Westmead Hospital, Sydney, NSW, Australia. [13]ICPMR, Department of Haematology, Westmead Hospital, Sydney, NSW, Australia. [14]Department of Haematology, Austin Health, Melbourne, VIC, Australia. [15]Cabrini Hospital, Melbourne, VIC, Australia. [16]Department of Haematology, Royal Adelaide Hospital, Adelaide, SA, Australia. [17]University of New South Wales, Sydney, NSW, Australia. [18]Royal North Shore Hospital, Sydney, NSW, Australia. [19]Haematology Research Group, Heart Research Institute, Sydney, NSW, Australia. [20]Institute of Haematology, Royal Prince Alfred Hospital, Sydney, NSW, Australia. [21]Medical Oncology Group, Liverpool Clinical School, School of Medicine, Western Sydney University and Ingham Institute for Applied Medical Research, Liverpool, NSW, Australia. [22]South-West Sydney Clinical Campuses, UNSW Medicine & Health, Sydney, NSW, Australia. [23]Sydney Medical School, Nepean Clinical School, Faculty of Medicine and Health, University of Sydney, Nepean Hospital, Kingswood, NSW, Australia. [24]Blacktown Hospital, Sydney, NSW, Australia. [25]Wollongong Hospital, Wollongong, NSW, Australia. [26]Liverpool Hospital, Sydney, NSW, Australia. [27]Central Coast Health, Gosford Hospital, Gosford, NSW, Australia. [28]St. George Hospital, Sydney, NSW, Australia. [29]Department of Haematology, Calvary Mater Hospital, Waratah, NSW, Australia. [30]University of Newcastle, Callaghan, NSW, Australia. [31]Precision Medicine Program, Hunter Cancer Research Institute, New Lambton Heights, NSW, Australia. [32]Otway Pharmaceutical Development and Consulting Pty Ltd, Forrest, VIC, Australia. [33]Scientia Clinical Research, Medical Oncology, Prince of Wales Hospital, Sydney, NSW, Australia. [34]School of Biomedical Sciences, The University of Hong Kong, Hong Kong, SAR, China. [35]Department of Clinical Haematology, Prince of Wales Hospital, Sydney, NSW, Australia. [36]School of Mathematics and Statistics, University of New South Wales, Sydney, NSW, Australia. ✉ e-mail: j.thoms@unsw.edu.au; mark.polizzotto@anu.edu.au; jpimanda@unsw.edu.au

