## [Transparent Peer Review file · Nature Communications]

Clinical Response to Azacitidine in MDS is Associated with Distinct DNA Methylation Changes in HSPCs

Corresponding Author: Professor John Pimanda

Version 0:

Reviewer comments:

Reviewer #1

(Remarks to the Author)

Hypomethylating agents, e.g., 5-azacytidine, are central to treating myeloid malignancies. Unfortunately, response rates are limited in magnitude and duration, and lessons around mechanism-of-action and resistance are needed to improve these response parameters. In this regard, this manuscript is an important addition to the literature, since it describes a prospective clinical trial which sought to investigate and better understand mechanism-of-action by rigorous flow cytometry and cell cycle analyses coupled with drug incorporation measurements by mass spectrometry, DNA methylation analyses by various methods, and mutation variant allele frequency analyses, an unprecedented combination of prospective analyses on serial clinical samples, providing a wealth of clinical sample derived data.

I have the following comments and recommendations:

1. Clinical response in myeloid malignancies is defined by myeloid blood count recovery, which in turn likely depends on the numbers of myeloid stem and progenitor cells in patient bone marrow, which likely varies from patient to patient driven by age, disease type, disease duration, previous treatments, etc.. However, this likely fundamental driver of response vs non-response has been addressed only in limited ways in previous studies (e.g, in reference #40 in the present manuscript). The flow cytometry and cell cycle analyses performed at baseline in the present study could provide further insights into the role of this baseline parameter, e.g., by including in Figure 1 graphs showing the proportion of HSC and HPC to other cells (non-CD34+) in responders vs non-responders.

- Also in this regard, marrow cellularity estimates by hematopathology have been traditionally used to estimate this parameter of marrow reserve, and it would be relevant to show in relation to the HSC and HPC percentage numbers also marrow cellularity estimates by hematopathology (again, the overall goal will be to provide insights into the role of marrow reserves of HSC and HPC, mutated and unmutated, in determining response vs non-response to therapy)

2. A potentially critical insight provided by this manuscript is a central role of drug incorporation into the DNA of myeloid cells and response, as shown in Fig 2E and 2F. These analyses could potentially provide an insight also into mechanisms-of-resistance: for example, in serial analyses in responding patients was a decrease in drug incorporation noted at time of relapse or was there no change in drug incorporation at relapse? These analyses would be especially valuable if the character of the relapse is described, e.g., with overt increase in myeloblasts or just by decline in blood counts with decreasing marrow cellularity. It would also be important to know that patients were continuing to receive cycles of therapy at time-of-relapse since this will obviously impact possibilities of drug incorporation.

3. CpG locus specific methylation inherently varies by cell lineage and maturation stage. In the discussion, it would be important to point out that differences in lineage and maturation composition of samples might impact interpretation of CpG locus specific methylation.

Reviewer #2

(Remarks to the Author)

Reviewer #3

(Remarks to the Author)

I co-reviewed this manuscript with one of the reviewers who provided the listed reports. This is part of the Nature

Communications initiative to facilitate training in peer review and to provide appropriate recognition for Early Career Researchers who co-review manuscripts.

Reviewer #4

(Remarks to the Author)

The manuscript by Thoms and colleagues entitled "Clinical Response to Azacytidine in Myelodysplastic Neoplasms is Associated with Distinct DNA methylation changes in haematopoietic stem and progenitor cells in vivo" presents an interesting clinical translational study of patients enrolled in a phase 2 trial for the use of AZA for MDS. Overall, the majority of the results are null, showing little differences between responders and non-responders. This may be due to the very small numbers in the study (n=16 responders vs n=10 non-responders). The most interesting finding was the shift from G0 to G1 in HSC and HPC cells in response to the AZA treatment although a comparison across responders vs non-responders is missing.

Comments on Text:

1. Statements like this should be avoided: "in HPCs there was a trend for responder patients to have an increased proportion of cells in S/G2/M at baseline (R: [5.0, (2.9-5.7)]; NR: [3.2, (2.2-6.0)]), although this did not reach statistical significance". Not significant can be interpreted as not different. there are several examples of this throughout the manuscript.
2. Stating a significant difference in Fig 2C in responders needs to be confirmed once the outliers are included. (it is likely they are not actually significant).
3. Conceptual issue to discuss - if patients had already been treated with injection before the oral AZA, could their response to oral AZA be blunted?
4. when you look for gene enrichment in the 18 CpGs that are hypomethylated long term, it seems an obvious analysis to also look at the 83 hypermethylated regions.
5. Fig 2C (ii) and (iii). although you show a difference within responders and not within non-responders. to strengthen this finding it would be important to compare the delta (change from D1 to D28) in responders vs non-responders to show that there is a significant difference between groups. it appears from the figure that there might not be a real difference.
6. In the discussion there is a bit of disjoint between the statement that "methylation and expression are concordant with biological processes underlying clinical response" vs the fact that you didn't see an association with clinical response.
7. the discussion is lacking a discussion of the limitations of the study (some are raised in the last paragraph), however, the main limitation is the small study size. A power calculation could have been performed to determine the effect sizes detectable with this cohort, or at the least use the existing data to propose a larger study that would be sufficiently powered to validate these findings.

Comments on Figures:

Fig 2 - important to state on this figure the number of responders and non-responders (is the difference in numbers the only reason one might be significant while the other not?). Also should state "ns (not significant)" where appropriate, as you do in Fig 3. Where are the outliers in Fig 2B(ii) and (iii) in the Cycle 1-6 figures, blue dot at 40% and blue dot at 70%. You should recalculate statistics with these patients included.

Fig 3 - (D) and (E) interesting that drug incorporation is significant at D8, but the DNA methylation is not significant at D8. in (I) you should label the x- and y- axes with the methods (AZA-MS and LINE-1 PCR), as it is not obvious which is which. There is also no reporting of the correlation coefficient or P-values here.

Fig 4 - (F) is quite small and hard to see, perhaps a better design would be to make HOXB1 and EPO the width of the figure on top of each other. labels for heatmap scales are needed (D) and (H). What are the side colours indicating in (H)

Fig 5 - (D) should indicate the analysis is D1 vs D8 in responders only

Fig 6 - (A) - if Transformed % methylation is representing M-values, it would help to define that in the figure legend. are these numbers equally significant if presented as beta-values? Beta-values would present easier to interpret for general readers. (D) should be $p=0.18$ rather than $p \leq 0.18$

Extended Data Fig 5: - "Title" should be changed.

Co-reviewer comments:

A study on the clinical response of azacitidine in MDS patients shows differential DNA methylation alterations in

hematopoietic stem and progenitor cells. The findings suggest that unique methylation patterns may indicate therapy response, enabling tailored MDS medication. This study supports azacitidine's efficacy and sheds light on its molecular processes, improving our understanding of MDS therapy responses. The study has rigorous study design with several drawbacks. Most concerning is the study's small sample size, which may limit generalizability and statistical power. The follow-up period may not be long enough to capture long-term therapy outcomes. The efficacy of azacitidine is questioned due to the lack of well-defined control groups. Patient heterogeneity, including disease type and prior therapy, may also introduce confounding factors. Potential bias in patient selection and limited exploration of the underlying biological mechanisms pose challenges. Finally, the study did not mention clearly about patient permission and genetic data handling, which are critical in clinical research. Enhancing clarity, discussing results more thoroughly, and emphasizing study innovation will improve the manuscript's quality and impact.

Major Issues

Methods

1. The writers did not consider other clinical factors, aside from treatment, that can influence methylation differences. Would it be better to analyze the confounding factors that also influence patient methylation?
2. Although the participants were defined as high-risk MDS, CMML, or low blast AML, and ineligible for allogeneic stem cell transplant or intensive chemotherapy, there may be different methylation baselines among these three disease groups that contribute to the level of methylation after receiving AZA. How will this potential be addressed?
3. Serious adverse events were recorded only during oral AZA cycles, potentially missing delayed or cumulative effects that occur post-treatment or in subsequent cycles. This is essential for evaluating the comprehensive safety profile of AZA.
4. Clinical responses were initially assessed by attending clinicians, which, despite independent adjudication, may introduce subjective bias or variability in assessment standards and they might lack full patient context, impacting their interpretation of clinical responses compared to those directly involved in patient care.
5. The number of ethical approvals from institutions should be included.

6. Study Design:

1. The open-label, phase 2 multicentre, investigator-initiated trial has limitations that could introduce bias, including a lack of blinding, absence of a control group (placebo or comparator), and investigator bias. Phase 2 trials primarily focus on efficacy and often lack long-term safety data, making it possible for rare adverse events to go undetected. Researchers should address these limitations to minimize bias and improve the reliability of results.
2. The eligibility criteria for patients were not defined comprehensively.
3. The sample size is not adequately justified. A power analysis should be included to support the chosen sample size, ensuring that the study is powered to detect meaningful differences.
4. The absence of a well-defined control group limits the ability to draw robust conclusions about the efficacy of Azacitidine. Including a placebo or historical control group would strengthen the findings.

7. In the "Isolation of DNA and RNA"

1. Using two different kits (BioBasic and Qiagen) for DNA/RNA isolation may introduce variability in yield, purity, or efficiency, potentially affecting downstream analyses and data consistency. Quantifying DNA/RNA with both a Nanodrop spectrophotometer and a Qubit fluorometer can lead to inconsistencies, as Nanodrop is less specific and may detect impurities, whereas Qubit provides more precise quantification of nucleic acids.

Discussions

8. What are plausible reasons for increased replication being associated with increased AZA incorporation?
9. The finding that global methylation in PBMC was similar in responders and non-responders suggests that this measurement may not adequately capture clinically relevant epigenetic changes. It may also indicate that focusing on global methylation as a response marker has limited utility, prompting consideration of more targeted or region-specific methylation profiling for future studies. Would targeting methylation profiling be more beneficial in measuring clinical response? If so, what specific CpG sites do you suggest?
10. The persistence of mutated alleles across treatment cycles, regardless of clinical response, highlights a limitation in AZA's effectiveness in eradicating mutated cell populations. This limitation might restrict the long-term clinical benefit of AZA if mutant clones remain or expand post-treatment. How does your research implicate suggestions for clinical practice using AZA for this disease?
11. Contrary to prior findings, baseline cell cycle features did not predict response, which may limit the use of cell cycle markers for patient stratification. Will you consider additional markers or combinatory approaches to enhance predictive accuracy for clinical outcomes?
12. The authors should reference similar studies that explore the relationship between DNA methylation and treatment response in MDS. There is also limited comparison with existing studies on Azacitidine. A more thorough discussion of how these findings align or contrast with previous research would enhance the manuscript's contribution to the field.
13. The discussion provides a reasonable interpretation of the results; however, it lacks a critical evaluation of the limitations of the study. Acknowledging potential biases or confounding factors would provide a more balanced view.

Minor Issues

Abstract

1. The abstract provides a general overview but lacks specific quantitative results that would help readers gauge the significance of the findings. Including key statistics or outcomes would enhance clarity.

Writing issues

2. Abbreviations should be clearly defined when first mentioned.

Introduction

3. While the introduction outlines the importance of the study, it would benefit from a more detailed discussion of existing literature on Azacitidine and its mechanisms of action to establish a stronger rationale for the study.
4. Before explaining that high-risk, transplant-ineligible patients are treated with hypomethylating agents (HMAs) like azacitidine (AZA), it is best to briefly introduce current therapies for myelodysplastic neoplasms and their limitations, which

led to HMA development.

5. The introduction should clearly define research gaps by mentioning prior literature relevant to this research.

Results

1. Results should be presented more clearly, as some key findings are buried in text and could be highlighted more effectively.

2. The conclusion should explicitly state the clinical implications of the findings, including how these results might influence treatment protocols or future research directions.

Conclusions

3. Suggestions for future research are minimal. The authors should propose specific areas for further investigation based on their findings.

Reviewer #5

(Remarks to the Author)

Reviewer #6

(Remarks to the Author)

In this manuscript, authors have evaluated clinical outcomes in patients with high-risk MDS, CMML, or low-blast AML when treated with azacitidine (AZA). The patients were treated with AZA injections followed by oral AZA in a phase 2 clinical trial. 16 of 24 patients responded post 6 cycles of AZA injection, with complete or partial remissions while others had stable or progressive disease. Cell cycle analysis has not shown any correlation between clinical response and the cell cycle status of hematopoietic stem cells (HSCs) or progenitor cells (HPCs). Although previous reports suggested that active cycling HSPCs might predict response, there is no such correlation in the current study. Responders had baseline hypomethylation in promoter regions and CpG islands. Although this correlation was again not significant, there was enrichment of pathways in blood development and lineage-specific gene regulation.

Further, the decitabine (DAC) incorporation and DNA demethylation did not directly correlate with clinical outcomes, though responders had slightly higher drug incorporation during the injection phase. The data presented here hinted at DNA demethylation in CD34+ HSPCs as a potential response marker, but may be indirectly related to clinical outcomes. Lastly, patients had fluctuating mutation burdens and these mutations persisted regardless of clinical outcomes or methylation changes.

Comments

1. The results presented here are from a relatively small sample size. Since it was a small sample size, some patients transitioning from responders to non-responders (or vice versa) after treatment could lead to biological variability.

2. In Fig. 2C the lack of significant differences in S/G2/M phase in HSPC proportions raises questions about the underlying biological assumption. The report captures a trend of increased cell cycling in HSPCs among responders; statistical significance is not consistent. Is it due to insufficient power or variability in the measure?

3. In Fig 3B, why the drug incorporation was high but it doesn't correlate with DNA methylation. In 3D, the drug incorporation is high during the injection phase, but the DNA methylation was also significantly high. These data don't correlate well. The difference in DAC incorporation between injection and oral phases highlights pharmacokinetic variations that could influence treatment design, as oral AZA provides sustained demethylation. Yet, the lack of correlation with clinical response limits the utility of DAC incorporation as a predictive biomarker, if any.

4. In Fig 4, the reliance on healthy HSPC data for comparison may not fully capture methylation dynamics in MDS patients. How valid are these comparisons? Are there any other reports in the field

5. The study reports CpG site methylation differences, but it doesn't fully clarify how these differences translate to gene expression patterns hematopoiesis. Does the splice isoforms have any correlation? If the gene expression patterns are not significant, I suggest authors perform an rMATS analysis. It may be worthwhile doing splicing analysis for genes involved in key hematopoietic pathways. Emerging reports suggest splicing alterations could impact drug response.

6. The hypomethylation was prevalent in responders, yet the analysis lacked significant findings in lineage-specific differentiation pathways. Again, does the splice isoforms hold key information regarding response?

7. The use of LINE-1 promoter methylation in CD34+ cells as a proxy for global methylation is well-conceived but may oversimplify the complex epigenetic dynamics in MDS/CMML. Examining locus-specific demethylation could yield more precise markers for response.

8. Is there a control group in the study? Because the small sample sizes limit the study's power to establish causative associations.

Reviewer #7

(Remarks to the Author)

Version 1:

Reviewer comments:

Reviewer #1

(Remarks to the Author)

The authors have responded constructively to my comments and recommendations.

Reviewer #2

(Remarks to the Author)

We thank the authors for responding to our questions.

We are aware of the issue and we are sorry for this unexpected work.

The authors replied to all my comments.

They did a great job and this has been very much appreciated.

Reviewer #4

(Remarks to the Author)

The authors have addressed the comments of all the reviewers to my satisfaction and the manuscript has improved.

Reviewer #7

(Remarks to the Author)

Thoms et al response to reviewer comments

Reviewer #1 (Remarks to the Author):

Hypomethylating agents, e.g., 5-azacytidine, are central to treating myeloid malignancies. Unfortunately, response rates are limited in magnitude and duration, and lessons around mechanism-of-action and resistance are needed to improve these response parameters. In this regard, this manuscript is an important addition to the literature, since it describes a prospective clinical trial which sought to investigate and better understand mechanism-of-action by rigorous flow cytometry and cell cycle analyses coupled with drug incorporation measurements by mass spectrometry, DNA methylation analyses by various methods, and mutation variant allele frequency analyses, an unprecedented combination of prospective analyses on serial clinical samples, providing a wealth of clinical sample derived data.

Thank you

I have the following comments and recommendations:

1. Clinical response in myeloid malignancies is defined by myeloid blood count recovery, which in turn likely depends on the numbers of myeloid stem and progenitor cells in patient bone marrow, which likely varies from patient to patient driven by age, disease type, disease duration, previous treatments, etc.. However, this likely fundamental driver of response vs non-response has been addressed only in limited ways in previous studies (e.g. in reference #40 in the present manuscript). The flow cytometry and cell cycle analyses performed at baseline in the present study could provide further insights into the role of this baseline parameter, e.g., by including in Figure 1 graphs showing the proportion of HSC and HPC to other cells (non-CD34+) in responders vs non-responders.

We thank the reviewer for this helpful suggestion. The proportion of BMA mononuclear cells which are immunophenotypically HSPCs (single/lineage negative/DAPI positive/CD34 positive) was extracted for each patient for whom both baseline cell cycle and clinical response data was available (n=26). Our antibody panel includes CD38 to allow discrimination of stem cells (CD38lo) and progenitor cells (CD38hi). However, CD38 staining has a continuous distribution in CD34pos HSPCs and thus an arbitrary cut off is used to distinguish cell types (~15% lowest CD38 expression is called as stem cells and the remainder as progenitors). This allows comparative analysis of cell cycle in those populations but does not add any discriminatory power in the suggested analysis since HSC/HPC are fixed proportion subsets of HSPCs. We have

therefore conducted our analysis on CD34pos HSPCs, noting that cell cycle measurements were performed on Ficoll separated cells that had undergone a freeze/thaw cycle which is likely to lead to depletion of some mononuclear cell types (eg/ neutrophils).

Taking all patients together, the non-responders had a slightly higher proportion of HSPCs in the bone marrow at diagnosis compared to patients who did respond (new Supplementary Fig. 1a; $P = 0.045$, unequal variances two-sample t-test, $n=26$).

- Also in this regard, marrow cellularity estimates by hematopathology have been traditionally used to estimate this parameter of marrow reserve, and it would be relevant to show in relation to the HSC and HPC percentage numbers also marrow cellularity estimates by hematopathology (again, the overall goal will be to provide insights into the role of marrow reserves of HSC and HPC, mutated and unmutated, in determining response vs non-response to therapy)

Bone marrows were scored as (1) Hypocellular; (2) Normocellular; (3) Hypercellular; or (4) Markedly hypercellular; based on standard of care clinical reporting of bone marrow trephines (BMT) at baseline. Corresponding BMT and flow cytometry immunophenotyping data were available for 27 patients, and of these, 25 had clinical response data available. BMT from most patients were scored as hypercellular (14/25) or markedly hypercellular (7/25), with the remainder scored as normocellular (2/25) or hypocellular (2/25), and there was a wide range of %HSPC in BMA in each class. We conducted a gamma test to assess whether the proportion of responders increases or decreases as cellularity increases ($\gamma = -0.7895$, $P = 0.003$), thus the proportion of responders decreases as cellularity increases (new Supplementary Fig. 1b left panel). However, we find no clear relationship between cellularity and %HSPC in BMA (right panel).

Data relating to these two questions has been added to the manuscript in two new figure panels, Supplementary Fig. 1a, 1b, and in new Supplementary Table 9, the manuscript text has been updated to read “The proportion of CD34+ HSPCs in the BM-MNC fraction was slightly elevated in non-responders (Supplementary Fig. 1a) and increased bone marrow trephine (BMT) cellularity was associated with a higher proportion of non-responders (Supplementary Fig. 1b, Supplementary Table 9).” (line 151-154), and methods updated to include additional statistical methods (line 607-611).

2. A potentially critical insight provided by this manuscript is a central role of drug

incorporation into the DNA of myeloid cells and response, as shown in Fig 2E and 2F. These analyses could potentially provide an insight also into mechanisms-of-resistance: for example, in serial analyses in responding patients was a decrease in drug incorporation noted at time of relapse or was there no change in drug incorporation at relapse? These analyses would be especially valuable if the character of the relapse is described, e.g., with overt increase in myeloblasts or just by decline in blood counts with decreasing marrow cellularity. It would also be important to know that patients were continuing to receive cycles of therapy at time-of-relapse since this will obviously impact possibilities of drug incorporation.

We thank the reviewer for this insightful question. For this analysis we focus on patients with clinical response after 6 treatment cycles where we can directly compare AZA incorporation kinetics at clinical response (cycle 7) and a subsequent timepoint where the patient continued to respond (n=5) or had progressed/relapsed (n=7). Of the progression/relapse patients, two had marked increases in blast counts to >30%, the remaining patients had declining peripheral counts, accompanied in some cases by more moderate increases in blast %. We compared maximal DAC incorporation into DNA during cycle 7, and in the last treatment cycle for each patient (either cycle 12, or the cycle immediately preceding progression/relapse).

Maximal DAC incorporation into DNA during cycle 7 (C7) and in the final treatment cycle (either cycle 12 or the cycle immediately preceding progression/relapse) in patients showing clinical response at cycle 7.

Overall, we note moderate changes in the degree of drug incorporation over time in multiple patients, but the direction of change does not correlate with progression/relapse. We do not observe any statistically significant differences between the response groups at cycle 7. This is consistent with overall incorporation data during the oral phase (Figure 3f). Most patients have measurable drug incorporation, and in

fact the highest incorporation levels were observed in patients who progressed/relapsed (note this trend was also noticeable in the entire data set for the oral phase (Figure 3f). Overall, we do not see evidence for significant changes in drug incorporation that might indicate that relapse is imminent. Full incorporation kinetics are provided in Supplemental Fig. 3 and 4 (which have been updated to show the start of each treatment cycle), and we now include peripheral counts and BMA blast counts for all patients in Supplementary Table 2.

3. CpG locus specific methylation inherently varies by cell lineage and maturation stage. In the discussion, it would be important to **point out that differences in lineage and maturation composition of samples** might impact interpretation of CpG locus specific methylation.

We thank the reviewer for this suggestion and have included an additional sentence in the discussion “: Since CpG methylation varies with cell lineage and maturation 78, differences in the CD34+ progenitor composition may well impact interpretation of the current data. ” (line 521-522)

Reviewer #3 (Remarks to the Author):

Thank you

Reviewer #4 (Remarks to the Author):

The manuscript by Thoms and colleagues entitled "Clinical Response to Azacytidine in Myelodysplastic Neoplasms is Associated with Distinct DNA methylation changes in haematopoietic stem and progenitor cells in vivo" presents an interesting clinical translational study of patients enrolled in a phase 2 trial for the use of AZA for MDS.

Thank you

Overall, the majority of the results are null, showing little differences between responders and non-responders.

Although there were no significant differences in baseline cycling (HSCs and HPCs), DAC incorporation in DNA, or *global* DNA methylation in PB and BM MNCs and HSPCs, as highlighted in the title of this report, there were *locus specific* CpG methylation differences in HSPCs in responders vs non-responders. There are currently no robust indicators of AZA responsiveness. As these baseline and early drug induce differences were shared amongst responders they have value as potential biomarkers of AZA responsiveness.

This may be due to the very small numbers in the study (n=16 responders vs n=10 non-responders).

Patient enrolment numbers for the oral AZA phase 2 trial was powered for its primary objective- differences in DAC incorporation and DNA methylation in responders vs non-responders during injection AZA and oral AZA. Based on DAC incorporation/DNA methylation measurements in AZA responder/non-responders (Unnikrishnan et al Leukemia 2018), 20 patients were calculated to yield sufficient power to give a standardised difference of 0.77 with 90% confidence (Trial protocol: section 6, page 26, now included in methods line 581-582). We enrolled 40 patients estimating that approximately 50% of patients commencing injection AZA are expected to reach end of treatment, with approximately 50% of these expected to respond. Of the 40 patients enrolled, 24 reached the first assessment timepoint (16 responders and 8 non-responders). 5 patients progressed prior to this timepoint and were classified as drug non-responders. The remaining patients were withdrawn from the study either by participant preference or changes in clinical goals (detailed in Figure 1c).

The most interesting finding was the shift from G0 to G1 in HSC and HPC cells in response to the AZA treatment although a comparison across responders vs non-responders is missing.

We have performed a revised statistical analysis to compare responders and non-

responders, please refer to detailed response to Q5 below (response to reviewers line 230-249).

Comments on Text:

1. Statements like this should be avoided: "in HPCs there was a trend for responder patients to have an increased proportion of cells in S/G2/M at baseline (R: [5.0, (2.9-5.7)]; NR: [3.2, (2.2-6.0)]), although this did not reach statistical significance". Not significant can be interpreted as not different. there are several examples of this throughout the manuscript.

We thank the reviewer for pointing out these ambiguous statements and have updated the manuscript as follows:

"Although RNA sequencing data suggested cell cycle differences between response groups at baseline (Supplementary Fig. 2a, 2b), the proportion of HPCs in S/G2/M at baseline were similar (Figure 2e; R: [5.0, (2.9-5.7)]; NR: [3.2, (2.2-6.0)]), and a published 20-gene signature did not dichotomise patient outcome in this cohort (Supplementary Fig. 2c)." (line 171 – 174).

"Responder patients showed an increase, and non-responder patients a decrease in the proportion of cells in S/G2/M over the oral phase (Supplementary Fig. 2h)." (line 193 – 194)

"Comparing CD34+ cells from responders and non-responders, average demethylation was greater in responders, most markedly at C7D22 (Figure 3j)." (line 252 – 254)

At C1D1, we observed minimal enrichment of pathways regulating myeloid and erythroid differentiation (Figure 4i), consistent with all patients being unwell at this timepoint." (line 309 – 311)

"Overall gene expression changes at C1D8 were dominated by cell cycle pathways, an expected effect of AZA treatment (Figure 5e), and response-specific enrichment of pathways associated with myeloid and erythroid differentiation which were shifted compared to C1D1 (Figure 5f)." (line 338 – 341)

2. Stating a significant difference in Fig 2C in responders needs to be confirmed once the outliers are included. (it is likely they are not actually significant).

We apologise for the lack of clarity on this point. The left hand panels in 2B (now Figure 2b, 2c, 2d; updated to match journal style) and 2C (now Figure 2e, 2f, 2g) show patients at the start of cycle 1 and include every patient for whom a clinical outcome is available. This includes patients who progressed during the first 6 cycles (ie/ are non-responders). The right-hand panel shows per patient comparisons between the start of cycle 1 and

the end of cycle 6 – as this is a paired analysis any patient with missing data at either time point cannot be shown on the plot. Reasons for missing data include progression prior to the end of cycle 6, and samples where the number of HSCs observed by flow cytometry was too low (<50 events) to reliably assign cell cycle phase percentages. This is now indicated in the figure legends eg/ “Lines indicate paired samples from a single patient, only patients with data at both timepoints are shown. “(line 878-879, line 886-887). The number of samples shown in each plot is now explicitly stated in the figure legend (line 800, line 888).

3. Conceptual issue to discuss - if patients had already been treated with injection before the oral AZA, could their response to oral AZA be blunted?

This is indeed possible. At the time of initiating the study, in contrast with injection AZA (used as standard of care since 2004), there was limited clinical response data for the investigational product- oral AZA (CC486) and no data for in vivo DAC incorporation or DNA demethylation. As such the HREC was not comfortable approving upfront treatment with oral AZA, instead requiring the final crossover trial design. We have added this possibility as a discussion point “In this study, all patients were treated first with injected and then oral AZA, so we cannot rule out the possibility that prior treatment with injection AZA modulated incorporation and demethylation kinetics during the oral phase.” (line 450 – 453)

4. When you look for gene enrichment in the 18 CpGs that are hypomethylated long term, it seems an obvious analysis to also look at the 83 hypermethylated regions.

We apologise that this result was not clearly described. Pathway enrichment analysis was performed on the hypermethylated regions, but no enriched pathways were identified. We have updated the text as follows: “Pathway analysis of genes associated with hypermethylated CpGs returned no hits. Eighteen CpGs were hypomethylated in responders (Supplementary Fig. 8a, 8b), and associated genes were enriched in pathways including ...” (line 374-376).

5. Fig 2C (ii) and (iii). although you show a difference within responders and not within non-responders. to strengthen this finding it would be important to compare the delta (change from D1 to D28) in responders vs non-responders to show that there is a significant difference between groups. it appears from the figure that there might not be a real difference.

We agree with the reviewer that the figure suggests that the changes in G0 and G1 from C1D1 to C6D28 might be similar in responders and non-responders. Indeed, we had

attempted to express this in the original text where we noted “...there was an overall decrease in the proportion of quiescent HPCs ... and an increase in the proportion of G1 cells both of which were significant only in responders” (line 180-183). In response to the reviewer’s pertinent suggestion we have performed a new analysis using a linear mixed model approach with subject-level random effects to account for within-subject variability. Using this more powerful approach we indeed observe a statistical significance in non-responders as well as responders. The relevant p values are now included on the graphs (Figure 2c, 2d, 2f, 2g). This analysis also compared deltas between responders and non-responders and as expected, we find no difference. The manuscript text has been adjusted to better articulate these findings (line 170, line 182-183). This updated analysis also led to us confirming differences in S/G2/M cell between the start of cycle 7 and end of cycle 12 (Supplementary Fig. 2h) which was already evident at the end of cycle 12 (Figure 2h).

6. In the discussion there is a bit of disjoint between the statement that "methylation and expression are concordant with biological processes underlying clinical response" vs the fact that you didn't see an association with clinical response.

The statement has been rephrased to indicate that “observed alterations in *locus specific* CpG methylation and gene expression during AZA treatment are *consistent* with biological processes underlying clinical response ” (line 495 – 497).

7. the discussion is lacking a discussion of the limitations of the study (some are raised in the last paragraph), **however, the main limitation is the small study size**. A power calculation could have been performed to determine the effect sizes detectable with this cohort, or at the least use the existing data to propose a larger study that would be sufficiently powered to validate these findings.

Please refer to our earlier response regarding study size and power calculation (response to reviewers line 144-156). We have revised the discussion to indicate that the locus specific methylation differences that were noted in HSPCs of responders vs non-responders should be validated in a larger study “Finally, the relatively small sample size of our cohort limits our ability to detect small differences, and in particular, the locus-specific methylation differences we observe should be validated in a larger study” (line 527 - 529). Our data can be used to calculate patient numbers that would be needed to power such a study.

Comments on Figures:

Reviewer 4 (main reviewer)

Fig 2 - important to state on this figure the number of responders and non-responders (is the difference in numbers the only reason one might be significant while the other not?). Also should state "ns (not significant)" where appropriate, as you do in Fig 3. Where are the outliers in Fig 2B(ii) and (iii) in the Cycle 1-6 figures, blue dot at 40% and blue dot at 70%. You should recalculate statistics with these patients included.

This figure has now been updated to indicate ns where appropriate. The number of responders and non-responders in each panel are now included – these have been added to the figure legend to avoid cluttering the figure panels (eg line 876, line 880). Please refer to responses to Q2 and Q5 (response to reviewers lines 192 – 206 and 230-249) regarding data availability for comparisons between timepoints and updated statistics.

Fig 3 - (D) and (E) interesting that drug incorporation is significant at D8, but the DNA methylation is not significant at D8. in (I) you should label the x- and y- axes with the methods (AZA-MS and LINE-1 PCR), as it is not obvious which is which. There is also no reporting of the correlation coefficient or P-values here.

We agree that the different kinetics of drug incorporation and demethylation that we observe during injection vs oral treatment cycles are interesting. We would hypothesise that sustained exposure to AZA during the oral cycles would increase the chance of individual cells being in an appropriate cell cycle phase to incorporate the drug, and this in turn would lead to sequential cell cycles where DNMT1 would be depleted and DNA demethylated. Thus we observe greater DNA demethylation at timepoints more removed from drug administration.

We agree with the reviewer that (Figure 3i) was somewhat ambiguous. The axes have now been re-labelled to “LINE-1 demethylation relative to C1D1 (LINE-1 PCR)” and “Genome methylation (%) relative to C1D1 (AZA-MS)” to emphasise not only the methodologies used, but also the different assay targets (LINE-1 TEs compared to the entire genome). Given that the two assays measure different targets with differing baseline methylation levels, we expect imperfect concordance between these assays, even though both targets are demethylated in response to AZA treatment. To make this clearer, we have removed the regression line and instead coloured data points by concordance (green) or discordance (purple). This makes it clear that samples called as demethylated by AZA-MS are also called as demethylated by LINE-1 PCR and validates our use of LINE-1 PCR as a proxy measure of DNA methylation in low abundance samples.

Fig 4 - (F) is quite small and hard to see, perhaps a better design would be to make HOXB1 and EPO the width of the figure on top of each other. Labels for heatmap scales are needed (D) and (H). What are the side colours indicating in (H)

We thank the reviewer for these suggestions. Figure 4f has now been made larger and heatmap scale labels added for 4d and 4h. The cluster colours in Figure 4h correspond to enriched pathways shown in Supplementary Figure 5j. This has now been made explicit in the figure legend “Cluster colors correspond to pathways shown in Supplementary Figure 5j” (line 965 - 966).

Fig 5 - (D) should indicate the analysis is D1 vs D8 in responders only

We apologise for the lack of clarity. We have added an explicit title to the figure and the figure legend has been updated to read “Enrichment of CpGs differentially methylated in responders at genomic regions....” (line 980).

Fig 6 - (A) - if Transformed % methylation is representing M-values, it would help to define that in the figure legend. are these numbers equally significant if presented as beta-values? Beta-values would present easier to interpret for general readers. (D) should be $p=0.18$ rather than $p\leq 0.18$

We thank the reviewer for these suggestions. The plots in (a) have now been altered to show beta values, which scale from 0 (unmethylated) to 1 (methylated). The P values remain equally (or more) significant when presented this way. We also added new panels b and e to better display methylation differences across time points (previously only presented in the figure legend). The P value in f (previously d) has also been updated, with non-significant p values now denoted as “ns” to maintain consistency with other figures.

Extended Data Fig 5: - "Title" should be changed.

We have added the title: “Differences in site-specific methylation in CD34+ HSPCs immediately following the first cycle of oral AZA”. Figures were relabelled to match journal style, this is now Supplementary Figure 8.

Thank you

Co-reviewer comments:

A study on the clinical response of azacitidine in MDS patients shows differential DNA methylation alterations in hematopoietic stem and progenitor cells. The findings suggest that unique methylation patterns may indicate therapy response, enabling tailored MDS medication. This study supports azacitidine's efficacy and sheds light on its molecular processes, improving our understanding of MDS therapy responses. The study has rigorous study design with several drawbacks. Most concerning is the study's **small sample size**, which may limit generalizability and statistical power.

The study was powered for the primary objective as detailed in response to Reviewer 4 (Response to reviewers: line 142-156). The statistical considerations are outlined in the trial protocol (Trial protocol: section 6, page 26) and are now explicitly mentioned in the methods section (line 581-582).

The follow-up period may not be long enough to capture long-term therapy outcomes.

The objective of this study was not to assess the efficacy of AZA nor to ascertain clinical outcomes but rather to establish laboratory parameters associated with drug response.

The efficacy of azacitidine is questioned due to the lack of well-defined control groups.

The efficacy of Azacitidine, a standard of care treatment for MDS, CMML, and low blast AML, is already well established compared to the control group who received best supportive care (Silverman et al J Clin Oncol 2002 PMID: 12011120, Fenaux et al Lancet Oncology 2009 PMID: 19230772). Given its known efficacy, it would have been unethical to withhold AZA to patients with HR-MDS/CMML/LB-AML. Indeed, such a design would not have been approved by a HREC.

Patient heterogeneity, including disease type and prior therapy, may also introduce confounding factors.

Please see inclusion/exclusion criteria in the manuscript methods “Patients were eligible for enrolment following a new diagnosis of HR-MDS (IPSS; intermediate-2 or high-risk), CMML (bone marrow [BM] blasts 10-29%), or LB-AML (20-30% blasts).” (

Reviewer 4 (co-reviewer)

line 542-544), in the clinicaltrials.gov entry (<https://clinicaltrials.gov/study/NCT03493646>), and trial protocol which was submitted alongside the manuscript (Trial protocol: section 8 page 30). Although MDS is indeed a molecularly heterogeneous disease, the prognostic scoring systems (IPSS/IPSS-R) were designed to capture such heterogeneity for both clinical use and comparisons of patient groups enrolled in clinical trials. All patients were scored by IPSS and IPSS-R and were treatment naïve. No other disease modifying agents were allowed.

All myeloid neoplasms are inherently heterogeneous but share morphological and genetic features. HR-MDS/LB-AML/CMML (as classified by WHO 2017 criteria; Khoury et al Leukemia 2017) are responsive to HMAs and are eligible for standard of care injection AZA therapy on the pharmaceuticals benefit scheme in Australia- a consideration for their inclusion in the oral AZA trial. It is pertinent that almost all clinical trials registered on clinicaltrials.gov, which investigate novel HMA therapies for HR-MDS include all three diagnostic entities given their morphological and genetic overlap.

Of the 40 patients enrolled, 31 were HR-MDS, 6 were LB- AML with MDS features and 3 were CMML. The distinction between HR-MDS and low blast AML with MDS features (as all our patients were) is based on the % of blasts and these numbers are fluid. The WHO 2017 criteria were used to classify patients (Trial protocol: appendix II, page 65 - 68) when HR-MDS was classified as LB-AML when blasts were $\geq 20\%$. Although these numbers were retained in the 2022 revisions (<https://tumourclassification.iarc.who.int/chapters/63>), the accompanying ICC classification (Arber et al Blood 2022) classified HR-MDS with 10-19% as MDS/AML and those $>20\%$ as AML. CMML, which is classified as an MDS/MPN shares genetic and morphological features with HR-MDS/LB-AML with MDS features but uniquely presents with a high monocyte count. Bearing in mind that such morphological distinctions also exist in AML (M4-myelomonocytic, M5- monocytic, M6- erythroblastic, M7- megakaryocytic), current clinical trial designs for AML would include all morphological subtypes as the features that unite them are greater than those that separate them and they show broadly similar responses to standard of care treatment. The exception is AML-M3 where standard of care and clinical responses are very different to non-M3 AML. Taken together, the trial design including these entities is in line with current clinical guidelines.

Potential bias in patient selection and limited exploration of the underlying biological mechanisms pose challenges.

There is no discernible bias. Patient eligibility for HMA therapy in Australia is based on strict diagnostic criteria. Patients were recruited from large referral hospitals with highly specialised haemato-oncology services. All bone marrow and blood reports were independently reviewed by trial monitors. Patient inclusion criteria and the governance structures are clearly outlined in the trials protocol and the manuscript methods (line 542-544, line 564-573).

Finally, the study did not mention clearly about patient permission and genetic data handling, which are critical in clinical research

Participants provided consent for the study including collection and storage of tissue and genetic testing (Trial protocol: Appendix X: Master Participant Information Sheet and Consent Form “PARTICIPANT CONSENT INFORMATION SHEET AND CONSENT FORM CLINICAL RESEARCH INCLUDING GENETIC TESTING AND COLLECTION/STORAGE OF HUMAN TISSUE”, page 80-97). We thank the reviewer for flagging that this was not clearly described and have updated the methods section to state “All participants provided written informed consent for clinical research including genetic testing and collection/storage of human tissue” (line 568-570).

Enhancing clarity, discussing results more thoroughly, and emphasizing study innovation will improve the manuscript's quality and impact.

As detailed throughout this rebuttal document we have endeavoured to improve clarity and better emphasise and discuss our discoveries.

Major Issues

Methods

1. The writers did not consider other clinical factors, aside from treatment, that can influence methylation differences. Would it be better to analyze the confounding factors that also influence patient methylation?

Clinical factors, other than the malignancy itself and therapies such as HMAs that are used to treat it, that may influence methylation differences are obscure. Inclusion to the oral AZA trial was restricted to treatment naïve HR-MDS/CMML/LB-AML patients (see inclusion/exclusion criteria for bone marrow/blood parameters). No disease modifying agents other than AZA were allowed. Although the study was not powered to investigate differences in methylation associated with specific MDS related genes- we show methylation and DAC incorporation data with respect to VAFs of the most mutated genes that were represented in our patient cohort (Figure 7c). Furthermore, our study seeks “confounding factors” (e.g. cell cycle status, baseline promoter methylation) that

might be found in pre-treatment data to predict response that will not emerge for 6 months.

2. Although the participants were defined as high-risk MDS, CMML, or low blast AML, and ineligible for allogeneic stem cell transplant or intensive chemotherapy, there may be different methylation baselines among these three disease groups that contribute to the level of methylation after receiving AZA. How will this potential be addressed?

HR-MDS, CMML and LB-AML are related entities on a continuum of disease phenotypes that will no doubt undergo further stratification as knowledge improves. These myeloid neoplasms not only share a cell of origin, morphological features (monocytosis is a defining feature of CMML) and mutational landscape but also similar responsiveness to AZA. In this study, LB-AML patients were restricted to those with MDS associated morphological/genetic changes. As such the only distinguishing characteristic was the BM blast %- <20 % (HR-MDS) and 20-29% (LB-AML). This arbitrary distinction continues to evolve with classification updates and HR-MDS/LB-AML are derived from hypermethylated somatically-mutated HSPCs that can respond to HMA therapy via combinations of HMA-induced differentiation or ablation and are essentially the same disease along a spectrum of progression.

Nonetheless, we have performed a sub-analysis for HR-MDS (n=11/18 total patients with baseline RRBS data, the only subgroup with sufficient samples for meaningful analysis) and find that global CpG methylation is still significantly lower in responders at baseline (t-test with Welch's correction). These findings have been added to Supplementary Figure 5b. Irrespective of baseline differences that segregate with arbitrary diagnostic cutoffs or inter-patient variation, our longitudinal comparisons are performed in paired samples, allowing us to observe methylation differences after treatment regardless of inter-patient variation at baseline.

3. Serious adverse events were recorded only during oral AZA cycles, potentially missing delayed or cumulative effects that occur post-treatment or in subsequent cycles. This is essential for evaluating the comprehensive safety profile of AZA.

We thank the reviewer for this pertinent question. Injection AZA has been used as standard of care for 20 years. Reporting and independent monitoring of AE/SAE for standard of care treatment was unnecessarily onerous for sites and was not mandated by the PSC. On the other hand, oral AZA was an investigational product, and one of the secondary objectives was to assess its safety profile in a trial context. All grade #4 haem and grade #3 non-haem adverse events across all treatment cycles were recorded and are reported in Supplementary Tables 6 , 7, 8. We have updated the text in the methods section to read "Serious adverse events (SAE) were recorded during all cycles

(Supplementary Table 6, Supplementary Table 7). SAEs reported during the oral AZA cycles were assessed by independent medical monitors.” (line 571-573). Furthermore, during the one year follow up period, SAEs leading to participant death were recorded and are now included in Supplementary Table 7.

4. Clinical responses were initially assessed by attending clinicians, which, despite independent adjudication, may introduce subjective bias or variability in assessment standards and they might lack full patient context, impacting their interpretation of clinical responses compared to those directly involved in patient care.

The rigour of pathological training and diagnostics in Australia (FRACP) that are maintained by an annual QAP, mitigates against such bias. All response assessments made by treating clinicians were independently verified by trial monitors and adjudicated by a panel where there was disagreement. Furthermore, when our cohort were retrospectively assessed using IWG2023 guidelines (Supplementary Table 4) only a single response (from SD to HI at C12D29) was reclassified. We therefore respectfully assert that there is no subjective bias in interpretation of clinical responses. However, for clarity we have added an additional sentence to the methods “Clinical response was assessed by treating clinicians using IWG2006 guidelines and independently verified by trial monitors and adjudicated by a panel where there was disagreement.” (line 548-550)

5. The number of ethical approvals from institutions should be included.

We thank the reviewer for noticing this omission. The ethical approval numbers have now been included in the revised manuscript “The protocol received ethical approval (Ref#: 17/295, 2019/ETH05009) from the South Eastern Sydney Local Health District Human Research Ethics Committee, and participating sites received Institution approval to conduct the trial prior to commencing recruitment” (line 565-568). We note that under the Australian human ethics framework ethics approval granted by a single committee applies to all participating sites.

6. Study Design

a) The open-label, phase 2 multicentre, investigator-initiated trial has limitations that could introduce bias, including a lack of blinding, absence of a control group (placebo or comparator), and investigator bias. Phase 2 trials primarily focus on efficacy and often lack long-term safety data, making it possible for rare adverse events to go undetected. **Researchers should address these limitations to minimize bias** and improve the reliability of results.

While the crossover trial design in this study precludes blinding to treatment groups, all raw data analyses were performed by scientists blinded to patient outcome. We have updated the methods section to make this more clear “Raw data for primary and secondary outcomes was acquired by individuals blinded to patient outcome.” (line 589-590).

While phase 2 trials may often focus on efficacy, the primary objective of this trial was to compare AZA incorporation into DNA following injection or oral AZA treatment. Indeed, the efficacy of AZA is well established and withholding treatment for a placebo group would be unethical (Response to reviewers: line 368-374). Furthermore, AZA has been standard of care for more than a decade, ample time for rare adverse events to be detected in the thousands of patients treated over that time.

Finally, this study does include comparator groups (treatment with injected AZA and treatment with oral AZA) which underpin the primary objectives.

b) The eligibility criteria for patients were not defined comprehensively.

Eligibility criteria were extensively detailed in the trial protocol which was included in the reviewer package. Furthermore, eligibility criteria are comprehensively defined in the ClinicalTrials.gov record for the study. For clarity, we have updated the text in the method section to read “Extended eligibility criteria are available at <https://clinicaltrials.gov/study/NCT03493646>.” (line 542-544).

c) The sample size is not adequately justified. A power analysis should be included to support the chosen sample size, ensuring that the study is powered to detect meaningful differences.

Patient enrolment numbers for the oral AZA phase 2 trial was powered for its primary objective- differences in DAC incorporation and DNA methylation in responders vs non-responders during injection AZA and oral AZA. Based on DAC incorporation/DNA methylation measurements in AZA responder/non-responders (Unnikrishnan et al Leukemia 2018), 20 patients were calculated to yield sufficient power to give a standardised difference of 0.77 with 90% confidence. (Trial protocol: section 6, page 26). We have added a sentence to the method section to make this clear “Power calculations indicated 90% power to detect a standardised difference of 0.77 with n=20.” (line 581 - 582).

d) The absence of a well-defined control group limits the ability to draw robust conclusions about the efficacy of Azacitidine. Including a placebo or historical control group would strengthen the findings.

The efficacy of Azacitidine, a standard of care treatment for MDS, CMML, and low blast AML, is already well established (Silverman et al J Clin Oncol 2002, Fenaux et al Lancet Oncology 2009) and including an untreated control group would be unethical. Samples from an historical placebo group are not available to perform our assays which require immediate treatment of samples with THU and biobanking of longitudinal bone marrow mononuclear cells. Our analyses are centred on comparing responder to non-responder patients, so inclusion of placebo controls, even if possible, would be of minimal value.

7. In the “Isolation of DNA and RNA” using two different kits (BioBasic and Qiagen) for DNA/RNA isolation may introduce variability in yield, purity, or efficiency, potentially affecting downstream analyses and data consistency. Quantifying DNA/RNA with both a Nanodrop spectrophotometer and a Qubit fluorometer can lead to inconsistencies, as Nanodrop is less specific and may detect impurities, whereas Qubit provides more precise quantification of nucleic acids.

We agree that the methods section was ambiguous in this regard. Different kits were used to extract DNA/RNA for different experimental procedures, but there was consistency in isolation and measurement methods for each procedure. To clarify we have updated the methods section “DNA and RNA were isolated from frozen cell pellets or thawed CD34+ cells using either an All-in-One DNA/RNA miniprep kit (BioBasic #BS88203) for mass spectrometry experiments or an AllPrep DNA/RNA mini kit (Qiagen #80204) for RRBS and RNAseq experiments. DNA was quantified using either a Nanodrop spectrophotometer (Thermo) for mass spectrometry DNA samples and RNA samples or a Qubit fluorometer (Thermo) using a Quant iT Qubit dsDNA HS Assay Kit (Invitrogen #Q32851) for RRBS DNA samples.” (lines 635-640).

Discussions

8. What are plausible reasons for increased replication being associated with increased AZA incorporation?

AZA acts by incorporation into DNA (and RNA); details of this process are included in Figure 3a and stated explicitly in the introduction “HMA incorporation into DNA is cell cycle dependent, and we have previously reported a correlation between cell cycle parameters at diagnosis and subsequent patient response” (line 80-82). Incorporation into DNA, which is the mechanistic basis for DNMT1 degradation and subsequent DNA demethylation, depends on the cell passaging through S-phase.

9. The finding that global methylation in PBMC was similar in responders and non-responders suggests that this measurement may not adequately capture clinically

relevant epigenetic changes. It may also indicate that focusing on global methylation as a response marker has limited utility, prompting consideration of more targeted or region-specific methylation profiling for future studies. Would targeting methylation profiling be more beneficial in measuring clinical response? If so, what specific CpG sites do you suggest?

We agree with the reviewer that our data indicates that global methylation in PBMC does not capture clinically relevant epigenetic changes, an observation that led us to look at site specific CpG specific methylation in CD34+ cells (RRBS analysis) which revealed response-specific regions that are differentially methylated across multiple patients. However, further validation in larger cohorts would be needed to develop a diagnostic/prognostic tool based on methylation at specific nucleotides. Nonetheless, we agree that targeted methylation profiling is a promising avenue for further research.

10. The persistence of mutated alleles across treatment cycles, regardless of clinical response, highlights a limitation in AZA's effectiveness in eradicating mutated cell populations. This limitation might restrict the long-term clinical benefit of AZA if mutant clones remain or expand post-treatment. How does your research implicate suggestions for clinical practice using AZA for this disease?

We and others have previously reported the persistence mutant alleles in multiple publications (Merlevede et al Nat Commun 2016, Unnikrishnan et al Cell Reports 2017, Schnegg Kaufmann et al Blood 2023) and highlighted that response to AZA does not require eradication of mutant alleles but rather increased haematopoietic potential in mutant alleles (Schnegg Kaufmann et al Blood 2023). Eradication of mutant cells may be attractive in theory, but not if all productive haematopoiesis in MDS is from mutant cells (Schnegg Kaufmann et al Blood 2023). AZA therapy is continued indefinitely in patients who derive a clinical benefit. Most patients progress on treatment, and AML with antecedent MDS generally has very poor outcome, potentially due to the persistence of mutated clones.

11. Contrary to prior findings, baseline cell cycle features did not predict response, which may limit the use of cell cycle markers for patient stratification. Will you consider additional markers or combinatory approaches to enhance predictive accuracy for clinical outcomes?

As indicated in this report, we believe that methylation differences at specific loci could be used as biomarkers of AZA responsiveness. Please also refer to response 9 (Response to reviewers: line 602-615).

12. The authors should reference similar studies that explore the relationship between

DNA methylation and treatment response in MDS. There is also limited comparison with existing studies on Azacitidine. A more thorough discussion of how these findings align or contrast with previous research would enhance the manuscript's contribution to the field.

We have added references to additional studies exploring the relationship between DNA methylation and response to HMAs (48, 49; line 98). Many of the relevant studies measure methylation changes at only a few candidate gene promoters and are not directly comparable to the comprehensive RRBS data presented in this manuscript. Nonetheless, we have included a number of comparisons to existing studies of azacitidine including “activation of young TEs has been linked to clinical response to AZA” (line 493-494); “cell cycle features in HSPCs have been linked to patient response” (line 498 – 499); “there is increasing evidence that highly mutated stem cells with altered differentiation capacity contribute to improved peripheral counts in responders” (line 507 – 509).

13. The discussion provides a reasonable interpretation of the results; however, it lacks a critical evaluation of the limitations of the study. Acknowledging potential biases or confounding factors would provide a more balanced view.

Key limitations of our study are outlined in the final paragraph (line 516 – 529).

Minor Issues

Abstract

1. The abstract provides a general overview but lacks specific quantitative results that would help readers gauge the significance of the findings. Including key statistics or outcomes would enhance clarity.

The abstract already includes key outcomes e.g. “CpG methylation differences in HSPCs may explain variable response to AZA” (line 64). Given the breadth of the manuscript, inclusion of detailed data or statistics would not be possible within prescribed limits (150 words) nor in keeping with the recommended style of this journal.

Writing issues

2. Abbreviations should be clearly defined when first mentioned.

We thank the reviewer for this observation. We have scanned the manuscript and added in missing abbreviations including LC-MS/MS, BMT, AML, LINE, SINE, LTR, ChIPseq.

Introduction

Reviewer 4 (co-reviewer)

3. While the introduction outlines the importance of the study, it would benefit from a more detailed discussion of existing literature on Azacitidine and its mechanisms of action to establish a stronger rationale for the study.

An entire paragraph in the introduction (lines 79 – 96) provides a detailed discussion of the existing literature on azacitidine and its mechanism of action and the dire need for better therapies is clearly stated “Clinical response is generally not apparent until after 4-6 treatment cycles, and for patients with primary or secondary resistance to HMAs, treatment options are limited to enrolment in relevant clinical trials or supportive care.” (line 74 – 77). In this context, we also provide strong rationale for the current study;

“the association between HMA incorporation and DNA hypomethylation, and the clinical efficacy of these drugs is unclear” (lines 97 – 98)

“HMA incorporation and DNA hypomethylation kinetics have not previously been studied over the time span where clinical response becomes evident” (lines 101 – 102)

“these parameters have not been assessed in parallel with cell cycle characteristics or changes in clonal composition during HMA treatment” (line 102 – 104)

In summary, the introduction already includes the information suggested by the reviewer.

4. Before explaining that high-risk, transplant-ineligible patients are treated with hypomethylating agents (HMAs) like azacitidine (AZA), it is best to briefly introduce current therapies for myelodysplastic neoplasms and their limitations, which led to HMA development.

As described in the abstract (line 53) and introduction (line 70-73), HMAs are currently the frontline therapy for patients with high risk MDS unsuitable for transplant; this has been the case for more than a decade. A small number of alternate therapies are available, but these are effective in only a subset of patients (e.g. lenalidomide for patients with del(5q)) and of minimal relevance to this manuscript.

5. The introduction should clearly define research gaps by mentioning prior literature relevant to this research.

The introduction extensively references the literature (49 citations in this section) to define research gaps. In particular, the section from lines 97 – 104 specifically mention key knowledge gaps, and lines 104 – 112 explain how this study addressed these.

Results

Reviewer 4 (co-reviewer)

1. Results should be presented more clearly, as some key findings are buried in text and could be highlighted more effectively.

We agree that not all our findings are extensively highlighted; word and figure limits impose constraints that are particularly challenging for comprehensive studies such as ours. However, we have endeavoured to present all data as clearly as possible and have made many improvements (detailed throughout this document) in the revised version.

2. The conclusion should explicitly state the clinical implications of the findings, including how these results might influence treatment protocols or future research directions.

The purpose of this study was to increase mechanistic understanding of factors determining response to HMAs, and as such does not have direct clinical implications. However, following validation work in a larger cohort (as suggested in the discussion, line 527 – 529) we believe that locus-specific methylation changes may have utility in predicting patient outcomes and monitoring response the HMA therapies.

Conclusions

3. Suggestions for future research are minimal. The authors should propose specific areas for further investigation based on their findings.

Along with the proposal for additional studies at single cell resolution ” these data provide a foundation for more granular investigations of DNA methylation and sub clonal heterogeneity at single cell resolution” (line 476 – 477) we now include suggestions for follow up studies focussed on locus specific methylation differences in a larger cohort “...in particular, the locus-specific methylation differences we observe should be validated in a larger study. “(line 528 – 529).

Reviewer #5 (Remarks to the Author):

Thank you

Reviewer #6 (Remarks to the Author):

In this manuscript, authors have evaluated clinical outcomes in patients with high-risk MDS, CMML, or low-blast AML when treated with azacitidine (AZA). The patients were treated with AZA injections followed by oral AZA in a phase 2 clinical trial. 16 of 24 patients responded post 6 cycles of AZA injection, with complete or partial remissions while others had stable or progressive disease. Cell cycle analysis has not shown any correlation between clinical response and the cell cycle status of hematopoietic stem cells (HSCs) or progenitor cells (HPCs). Although previous reports suggested that active cycling HSPCs might predict response, there is no such correlation in the current study. Responders had baseline hypomethylation in promoter regions and CpG islands. Although this correlation was again not significant, there was enrichment of pathways in blood development and lineage-specific gene regulation. Further, the decitabine (DAC) incorporation and DNA demethylation did not directly correlate with clinical outcomes, though responders had slightly higher drug incorporation during the injection phase. The data presented here hinted at DNA demethylation in CD34+ HSPCs as a potential response marker, but may be indirectly related to clinical outcomes. Lastly, patients had fluctuating mutation burdens and these mutations persisted regardless of clinical outcomes or methylation changes.

Comments

1. The results presented here are from **a relatively small sample size**. Since it was a small sample size, some patients transitioning from responders to non-responders (or vice versa) after treatment could lead to biological variability.

We acknowledge that the sample size may appear to be relatively small, but the study was sufficiently powered to meet our primary and secondary objectives. We have added a sentence to the method section to make this clearer “Power calculations indicated 90% power to detect a standardised difference of 0.77 with n=20.” (line 581 - 582).

We agree that disease course differs between patients (evident in Figure 1C which tracks all patients across the trial). Realistically it is impossible to account for all confounding biological parameters in a human cohort. However, clinical response was assessed at two timepoints throughout the study and thus a nuanced response status can be confidently assigned to longitudinal samples. Changes in response status over time have also been taken into account in the analyses. For example, a patient classified as a responder at the C6D28 assessment point is grouped with the responders for analysis of data pertaining to that phase. However, if that patient subsequently transitioned to non-responder during the oral treatment phase (eg/ P04, P05) they were classed as a non-responder for analysis of data pertaining to that phase.

To make this clearer, we have amended the methods section to read “For analysis purposes, patients were classified by their response in the corresponding phase. For example, P10 was grouped with responders for the injection phase and non-responders for the oral phase (Figure 1c).” (line 550-552).

2. In Fig. 2C the lack of significant differences in S/G2/M phase in HSPC proportions raises questions about the underlying biological assumption. The report captures a trend of increased cell cycling in HSPCs among responders; statistical significance is not consistent. Is it due to insufficient power or variability in the measure?

In responding to another reviewer query (Response to reviewers: line 230-249) we have performed a new analysis of the data in Figure 2c using a linear mixed model approach with subject-level random effects to account for within-subject variability. Using this more powerful approach there are still no differences in S/G2/M proportion, however the increase in cell cycling (shift from G0 to G1) in HSPCs is consistently significant in responders and non-responders. The manuscript text has been adjusted to better articulate these findings (line 170, line 182-183).

Regarding the lack of difference in S/G2/M proportion, this cell cycle phase is of particular relevance for AZA treatment since drug incorporation into DNA necessarily occurs during S phase. The hope had been that baseline cell cycle parameters might prove a practical prognostic indicator of patients unlikely to respond to HMA therapy who might benefit from enrolment in clinical trials. Our previous study with smaller numbers of participants (Unnikrishnan et al 2017 Cell Reports) and indeed an interim analysis of this cohort (Unnikrishnan et al Blood 2019, vol 134(Supplement 1) p 4247) showed significant differences in S/G2/M which was lost as more patients were recruited to the study. While the current cohort is potentially statistically underpowered (power calculations were for the primary objective; cell cycle analysis was a secondary objective), our data suggests that the proportion of HSPCs in S/G2/M sits in a fairly narrow range for all patients (albeit with a few outliers). Even if a significant difference were to emerge in a much larger cohort it is unlikely that this finding would be clinically actionable given the large overlap between patients who go on to respond and those who do not.

3. In Fig 3B, why the drug incorporation was high but it doesn't correlate with DNA methylation. In 3D, the drug incorporation is high during the injection phase, but the DNA methylation was also significantly high. These data don't correlate well. The difference in DAC incorporation between injection and oral phases highlights pharmacokinetic variations that could influence treatment design, as oral AZA provides

sustained demethylation. Yet, the lack of correlation with clinical response limits the utility of DAC incorporation as a predictive biomarker, if any.

Figure 3b/3c show responder vs non-responder patients during the injection phase. Total AZA incorporation (3b) is observed in responders and non-responders but is higher in the patients who respond (and highest at Day 8 immediately following an injection cycle). Demethylation on the other hand (3c) is observed in both responders and non-responders (the dotted line at 100 indicates baseline global methylation level). Figure 3d and 3e provide a comparison of injection regimen (7/28 days) and the oral regimen (21/28 days). Although the highest amount of drug incorporation is observed during injection treatment, overall demethylation is greater in the oral phase which we attribute to the more sustained drug exposure during the oral phase. Overall, this data suggests that AZA-induced demethylation does not have a linear relationship with drug incorporation over a wide range of concentrations (perhaps maximal DNMT1 depletion is achieved relatively quickly after AZA incorporation), and that extended exposure to a lower dose of AZA could lead to greater global demethylation in all patients.

We agree with the reviewer that DAC incorporation in DNA and related global demethylation were poor correlates of clinical response and a key finding from this study was that these measures cannot be used as reliable biomarkers. However, locus specific CpG methylation in HSPCs did correlate with clinical response and could be used as biomarkers for clinical response, as such sites were shared between responders and not in non-responders.

4. In Fig 4, the reliance on healthy HSPC data for comparison may not fully capture methylation dynamics in MDS patients. How valid are these comparisons? Are there any other reports in the field?

CpG methylation in primary MDS HSPCs was assessed by RRBS, and assessment of differential CpG methylation was not reliant on healthy HSPC data; rather comparisons were made between response groups and across treatment timepoints. Many of the identified differentially methylated CpGs (DMC) were outside of promoter regions, making assignment of associated genes a challenge. We therefore used HiChIP data from healthy HSPCs to determine whether differentially methylated regions mapped to potential gene regulatory regions and were linked to specific genes (in healthy HSPCs). The reviewer is correct that these regions may or may not be used as regulatory elements in MDS - a point that we now acknowledge in the text "Many regulatory elements used in healthy HSPCs are likely to have regulatory functions in MDS HSPCs, although these may be modulated by network rewiring in transformed cells" (line 290-292). Additionally, mutant cells may generate new GREs in addition to those used by healthy HSCs in leukemia (Mulet-Lazaro and Delwel Blood Cancer Discovery 2024).

However, in contrast with AML blasts, MDS HSCs are able to differentiate (albeit with reduced efficiency) and produce circulating mature cells, and the mutant allele burden in mature cells reflects that of corresponding HSCs (Schnegg Kaufmann et al 2023). As such, MDS HSCs (in contrast to AML blasts) are very likely to use GREs that are used by healthy HSCs. To our knowledge, no HiChIP data exists for primary MDS HSPCs, and generating such data is currently technically challenging since experimental evaluation of GREs and chromatin contacts in MDS HSCs would require a minimum of 5×10^6 cells per sample - numbers that may be feasible in AML with high blast fractions but not in MDS. Regardless, our strategy links DMCs to specific genes based on actual looping structures in blood progenitors which we consider superior to the alternative which is to assign distal elements to the nearest gene.

5. The study reports CpG site methylation differences, but it doesn't fully clarify how these differences translate to gene expression patterns hematopoiesis. Does the splice isoforms have any correlation? If the gene expression patterns are not significant, I suggest authors perform an rMATS analysis. It may be worthwhile doing splicing analysis for genes involved in key hematopoietic pathways. Emerging reports suggest splicing alterations could impact drug response.

This comment is addressed in the response to Question 6 below.

6. The hypomethylation was prevalent in responders, yet the analysis lacked significant findings in lineage-specific differentiation pathways. Again, does the splice isoforms hold key information regarding response?

Although genes hypomethylated in responders at baseline were not enriched for pathways related to haematopoietic differentiation (Figure 4e), the additional hypomethylation observed in responders after a single AZA cycle were enriched for myeloid and leukocyte differentiation (Figure 5c) which might translate to increased differentiation potential in the treated cells. We agree with the reviewer that the link between changes in CpG methylation and gene expression patterns do not appear to be straightforward and thank them for their two comments suggesting we investigate RNA splicing. We first investigated differential transcript expression (DTE) at baseline and throughout the course of treatment (new Supplementary Figure 9a). DTE was observed at multiple timepoints, most prominently in responders at C7D22 v C7D1, and a small subset of DTE events occurred in key haematopoietic genes. We also investigated differential transcript usage (DTU) over the same comparisons. DTU events were again prominent in responders at C7D22 v C7D1 and also at baseline where such transcripts were more abundant in non-responders compared to responders (new Supplementary Figure 9b). However, similar to DTE, there was modest overlap with key haematopoietic genes. We next asked whether DTE or DTU events occurred at genes with differentially

methyated CpGs (DMC: new Supplementary Figure 9c). Overlap was generally restricted to DTE (except at baseline) and was most prominent at early time points where DMC were abundant.

Given that splicing alterations are a known feature of MDS, we find differential methylation and isoform usage of *SRSF5* particularly intriguing. We extended our DTE and DTU findings with an rMATS analysis to investigate novel splicing events before and after HMA treatment. At baseline, the skipped exon (SE) events were the most abundant, with a slightly higher number of unique events observed in responders. Retained intron events (RI) were less abundant but were highly skewed to non-responders. Gene ontology analysis of RI and SE transcripts identified the same enriched pathways in responders and non-responders; these were overwhelmingly RNA splicing and RNA processing related pathways (new Supplemental Figure 9d). The fact that pathways overlap at baseline (when all patients are unwell) suggests that the altered RNA splicing and processing pathways might be associated with disease pathophysiology. At C7D1, there were fewer unique RI events, and the imbalance between responders and non-responders had resolved (new Supplemental Figure 9e). However, there were striking differences in the pathways associated with RI events, with mostly viral response-related pathways enriched in responders, but RNA processing pathways enriched in non-responders. Pathways associated with SE events remained similar between responders and non-responders, although these were not the same pathways observed at baseline and might represent universal consequences of HMA treatment.

Overall, our analysis suggests that splicing differences, and RI events in particular, may be involved in HMA response, and this may be mediated in part by altered methylation. We have included this material in Supplementary Figure 9 (described in lines 381-401). However, we believe that developing these findings for translational use will require significant additional analysis and experimental validation which is beyond the scope of this manuscript.

7. The use of LINE-1 promoter methylation in CD34+ cells as a proxy for global methylation is well-conceived but may oversimplify the complex epigenetic dynamics in MDS/CMML. Examining locus-specific demethylation could yield more precise markers for response.

LINE-1 promoter demethylation qPCR was used when material was too limited to perform genome-wide demethylation assays by mass spectrometry (eg/ for CD34+ cells in Figure 3j). We agree that using LINE-1 methylation as a proxy for global methylation may oversimplify complex epigenetic dynamics, which is why we followed up LINE-1 methylation analysis in CD34+ cells with RRBS. While our single nucleotide resolution data revealed response-specific regions that are differentially methylated across multiple patients, further validation in larger cohorts would be needed to develop a

diagnostic/prognostic tool based on methylation at specific nucleotides. Furthermore, access to CD34+ cells for diagnostic/prognostic testing would be challenging in a routine clinical setting, and assays that use whole mononuclear cells (MNCs) from peripheral blood or bone marrow would be more amenable to clinical translation. We would predict that at least some prognostically significant CpGs in CD34+ cells would be shared in MNCs (these are the progeny of the CD34+ HSPCs), however confirmation would require matched genome-wide methylation data from CD34+ cells and MNCs and is beyond the scope of this study.

8. Is there a control group in the study? Because the small sample sizes limit the study's power to establish causative associations.

The efficacy of injection Azacitidine, a standard of care treatment for MDS, CMML, and low blast AML, is already well established (Silverman et al J Clin Oncol 2002 PMID: 12011120, Fenaux et al Lancet Oncology 2009 PMID: 19230772). Given its known superiority to best supportive care, it would have been unethical to withhold AZA from these patients and such a design would not have been approved by a HREC. However, our longitudinal study design facilitated paired comparisons within individuals which can reduce variability due to patient-to-patient heterogeneity and increase statistical power. Furthermore, the study was sufficiently powered to meet our primary endpoints (Trial protocol: section 6 page 26). A description of the anticipated study has now been included in the methods "Power calculations indicated 90% power to detect a standardised difference of 0.77 with n=20." (line 581 - 582).

Reviewer #7 (Remarks to the Author):

Thank you

Thoms et al response to reviewer comments

Reviewer #1 (Remarks to the Author):

The authors have responded constructively to my comments and recommendations.

Thank you

Reviewer #2 (Remarks to the Author):

We thanks the authors for responding to our questions.

We are aware of the issue and we are sorry for this not expxepted work.

The authors replayed all my comments.

They did a great job and this has been very much appreciated.

Thank you

Reviewer #4 (Remarks to the Author):

the authors have addressed the comments of all the reviewers to my satisfaction and the manuscript has improved.

Thank you

Reviewer #7 (Remarks to the Author):

Thank you